# The effects of forest canopy shading and turbulence on boundary layer ozone

P.A. Makar[1], R.M. Staebler[2], A. Akingunola[1], J. Zhang[1], C. McLinden[1], S.K. Kharol[1], B. Pabla[1], P. Cheung[1] & Q. Zheng[1]

The chemistry of the Earth's atmosphere close to the surface is known to be strongly influenced by vegetation. However, two critical aspects of the forest environment have been neglected in the description of the large-scale influence of forests on air pollution: the reduction of photolysis reaction rates and the modification of vertical transport due to the presence of foliage. Here we show that foliage shading and foliage-modified vertical diffusion have a profound influence on atmospheric chemistry, both at the Earth's surface and extending throughout the atmospheric boundary layer. The absence of these processes in three-dimensional models may account for 59–72% of the positive bias in North American surface ozone forecasts, and up to 97% of the bias in forested regions within the continent. These processes are shown to have similar or greater influence on surface ozone levels as climate change and current emissions policy scenario simulations.

[1] Air Quality Modelling and Integration Research Unit, Atmospheric Science and Technology Directorate, Environment and Climate Change Canada, 4905 Dufferin Street, Toronto, Ontario, Canada M3H 5T4. [2] Air Quality Processes Research Unit, Atmospheric Science and Technology Directorate, Environment and Climate Change Canada, 4905 Dufferin Street, Toronto, Ontario, Canada M3H 5T4. Correspondence and requests for materials should be addressed to P.A.M. (email: paul.makar@canada.ca).

The chemistry of the Earth's atmosphere close to the surface is known to be strongly influenced by the presence, absence, and type of vegetation, via hydrocarbon emissions[1,2] and deposition to the foliage[3,4]. These factors are common components of the computational models of atmospheric chemistry[5–8], which are in turn used for air pollution forecasting for current and future climates, and to formulate public policy with respect to regulation[9]. Other processes of importance in the atmosphere (transport, dispersion, gas and particle chemical reactions, emissions from human activities and particle microphysics) are of acknowledged importance and are accordingly represented in these models.

One of the key chemical species predicted by models of atmospheric chemistry is tropospheric ozone, due to its negative impacts on human health[10], the ecosystem, and crop production[11]. However, evaluations of these models[5,7,12] through the use of ozone monitoring network data[7,12] reveal they consistently over-predict surface ozone levels. This high bias is a long-standing issue that has persisted despite improvements to virtually all model components. For example, multiple model simulations of summer ozone across North America maintain positive biases of 5–12 p.p.b.v. throughout the diurnal cycle[7]. Locally, positive biases may be much larger, up 20 p.p.b.v. during the summer months[5]. An example of the spatial distribution of ozone biases for the month of July for North America is shown in Fig. 1a, using simulations carried out with version 2.1 of Environment and Climate Change Canada's Global Environmental Multiscale-Modelling Air-quality and Chemistry (GEM-MACH) air pollution forecast model[6–8,13]. The biases maximize over the eastern half of the continent, and along the California coastline. Multi-model ensembles[7,12,14] show that this is a common problem, with North American normalized mean biases (Fig. 1b) ranging from −10 to +26% and usually positive.

Most models and their ensemble show larger local positive biases, particularly when the data are examined regionally[14]. Figure 1c shows the bias performance of an ensemble of models for the regions outlined in black dash lines in Fig. 1a; the largest biases occur in the central to eastern sections of North America. Seasonally, these positive biases are the most pronounced in the summer[14]. While ozone biases in winter have been linked to

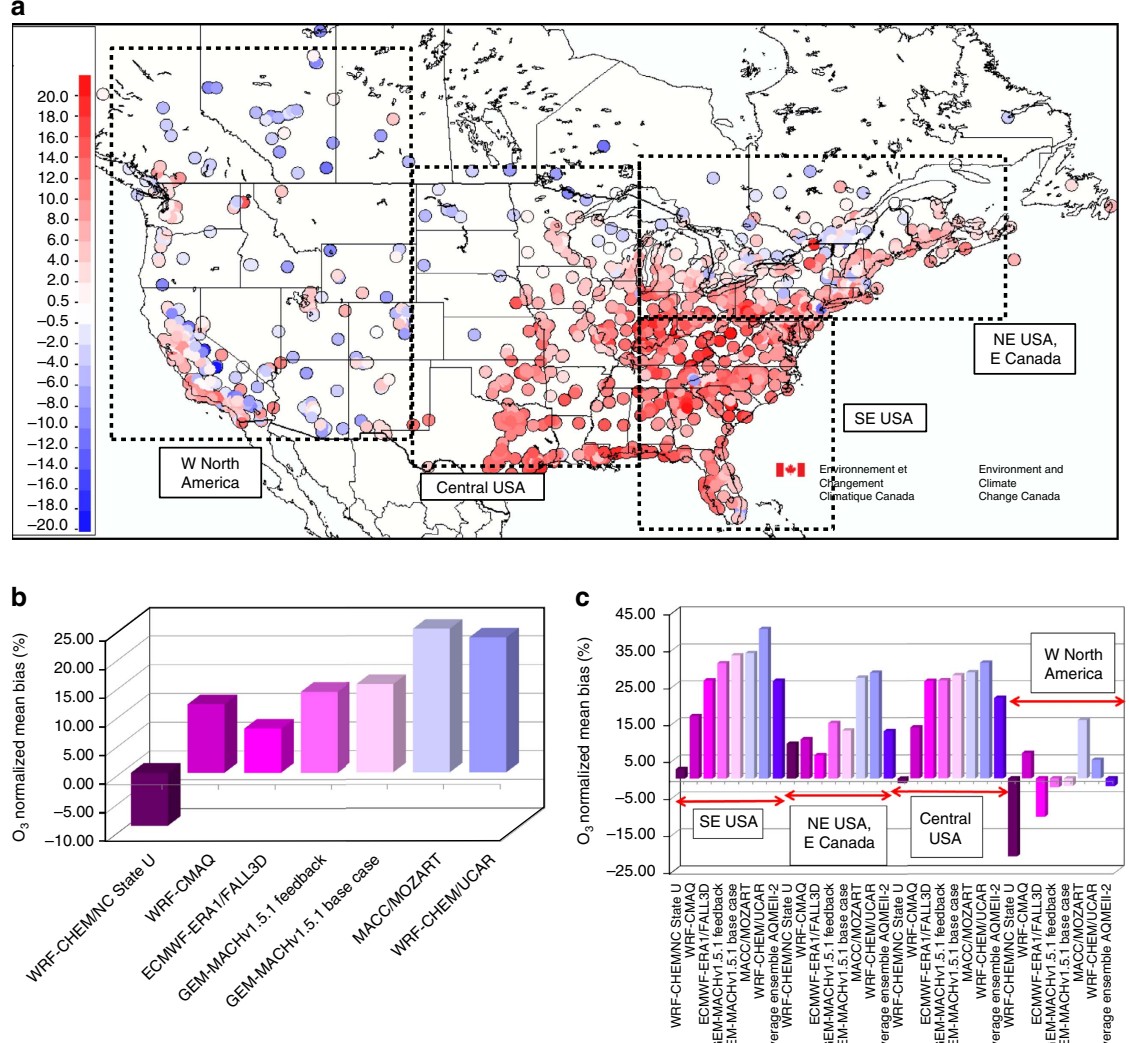

**Figure 1 | Typical biases in surface ozone predictions for North America.** (**a**) Average ozone bias at surface monitoring network observation stations, using the unmodified (base case) GEM-MACHv2.1 regional chemical transport model, for the month of July, 2010 (p.p.b.v.). Red regions indicate positive ozone biases, blue, negative ozone biases. (**b**) Normalized mean biases in ozone for North America for previous multiple models for July, 2010 (ref. 14) (p.p.b.v.). (**c**) As in **b**, subdivided into the regions outlined in **a**, along with the multi-model mean ensemble.

transcontinental transport[15,16], the larger summer bias within continents indicates missing processes as opposed to transport from abroad.

Parallel to these modelling efforts, measurements of ozone concentration in the forest research community have shown pronounced differences with height in dense forest canopies, suggesting a very different chemical environment exists above versus below the foliage[17–19]. These observations are consistent across very different forest environments. Measurements in the Amazon rain forest[17] indicate an 80% decrease in ozone from above the forest (79 m) to near-ground (0.5 m). Measurements and one-dimensional modelling for a forested site in central Massachusetts[18] showed decreased ozone of more than a factor of two going from above to below canopy. Past observations from the Borden Forest Research Station[19] showed a decrease in average ozone on the order 10–15 p.p.b.v. from above the canopy (42 m) to surface (1.5 m), with the greatest differences occurring in the summer months, and the smallest differences in the winter.

The sequence of reactions leading to the production of ozone is critically dependent on light intensity and wavelength[20]. If the light intensity becomes sufficiently low, ozone formation is inhibited due to lower formation rates of the hydroxyl radical and lower conversion rates of nitrogen dioxide to nitrogen oxide via photolysis. In addition, increases in stability near the surface of the Earth at night may allow the build-up of chemicals favouring ozone destruction (via, for example, 'titration' reactions with nitrogen oxides and/or double-bonded hydrocarbons). Reductions in light levels associated with foliage[21,22] are thus one possible cause for the observed decreases in ozone below foliage, consistent with known ozone chemistry. In addition, observations of forest canopy turbulence at multiple sites[23,24] suggest that vertical diffusivity in the under-canopy region will be greatly reduced, partially decoupling this region from the rest of the atmosphere, and increasing the relative influence of chemistry on ozone formation and destruction below the foliage. The reduced diffusivity may slow the upward transport of surface-emitted species such as nitric oxide, and hydrocarbons such as alkenes, increasing their concentrations in the below-canopy space. In a well-lit environment, higher concentrations of these species would lead to ozone formation (the dominant reactions being the conversion of nitric oxide to nitrogen dioxide via oxidation reactions, followed by nitrogen dioxide photolysis creating triplet-state monatomic oxygen, in turn biasing the balance of reactions towards ozone formation). However, in the darkened environment below the foliage, the dominant reactions are those of ozone 'destruction' through 'titration'; the reaction of nitric oxide and/or alkenes with ozone itself. Large Eddy

simulation (LES) studies with passive (that is, non-reactive) tracers[25,26] have demonstrated these 'trapping' effects of forests, as well as a third important process: species emitted in 'non-forested' regions will collect and be trapped in adjacent downwind 'forested' regions. This 'collection' effect may further enhance the below-canopy pool of precursors to ozone destruction, in the darkened below-canopy environment. These combined effects may result in a shift of photochemical regime away from ozone production and/or towards enhanced ozone destruction.

Here we show that tropospheric ozone is significantly reduced through the combined effects of forest canopy shading and turbulence. About one-third of the reductions can be attributed to forest canopy shading alone, with an additional two-third of the net reduction attributable to the forests' reduction in turbulence relative to non-forested environments. The inclusion of these processes largely corrects the long-standing positive bias in forecasts of North American surface-level ozone, and may account for a known deficiency in the data assimilation of satellite-derived ozone columns.

## Results

**Forest canopy ozone formation and removal.** Our investigation began with a re-examination of our multi-year record of observations within Borden Forest, at fixed heights throughout the canopy[19]. Ozone data were aggregated by season at the observation heights; these demonstrate the decrease in average ozone below the foliage seen in other studies (Fig. 2a). We found that the fraction of above-canopy photosynthetic photon flux density (PPFD), transmitted through the foliage to the ground, minimized in the summer months (Fig. 2b,c). The cause of this reduction in light intensity is the increased attenuation of light due to the presence of the leaves from deciduous vegetation, present only from spring through fall. Our observations thus demonstrate the co-location of the reduction in light intensity with the drop in ozone concentrations noted above.

The multi-model simulations of Fig. 1 and observation data shown in Fig. 2 led us to consider the potential influence of foliage shading and turbulence on larger scale ozone formation and destruction, possibly accounting for the positive biases shown in Fig. 1, and present in many other regional model simulations[5,12,14]. We hypothesize that, relative to non-vegetated surfaces, foliage-induced attenuation of downward radiation, combined with foliage-induced modifications to vertical diffusivity, suppress daytime photochemistry and limit the vertical mixing of chemicals emitted near the Earth's surface. These combined effects result in a shift of photochemical regime

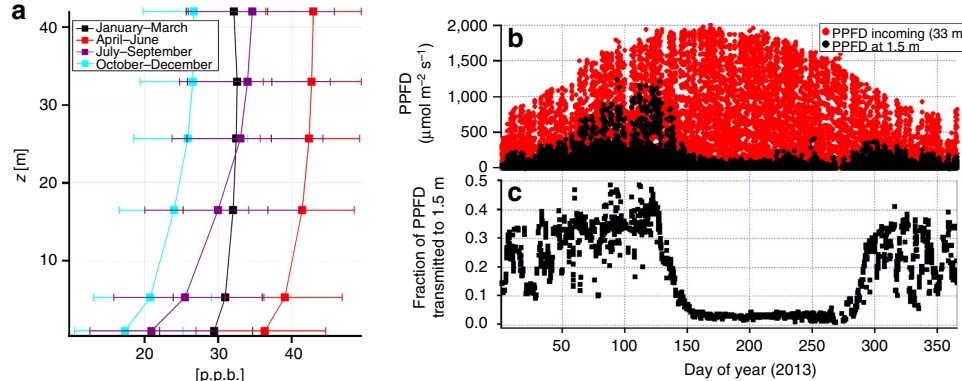

**Figure 2 | Long-term observations of ozone and PPFD at Borden Research Station.** (**a**) Four-year seasonal average of ozone concentrations measured at different heights in the canopy (key indicates the averaged months of the year with interquartile ranges); (**b**) PPFD above the canopy (33 m) and near the surface (1.5 m) during 2013; (**c**) fraction of above canopy PPFD transmitted to the ground during 2013.

away from ozone production and/or towards enhanced ozone destruction.

**Global forest canopy satellite data.** Satellite-data-derived images of canopy height and vertically integrated leaf area per unit ground area (Fig. 3a,b) provide some support for the hypothesis, in that the regions with the deepest forests (Fig. 3a) and thickest foliage (Fig. 3b) correspond reasonably well to the locations of positive ozone biases in Fig. 1a. Multi-model ensembles of European ozone predictions[12,14] compared to the same global forest canopy satellite data have a similar correspondence between positive ozone biases and forest locations (not shown). The location of deep forests across the world (Fig. 3c) suggests that forest processes may have a significant impact on global near-surface atmospheric chemistry.

**Spatial representativeness of air quality models.** We note that a key issue for air quality models is that of spatial representativeness—the ability of a model to distinguish small features in the domain is limited by the available computation resources, and current large-domain air-quality models have resolutions typically on the order of 2.5–30 km. As a result of these resolution limitations, observational data sets, which are collected at specific points in space, must be compared to average conditions over regions between $2.5 \times 2.5$ and $30 \times 30$ km$^2$. The extent to which the observations and model values match will thus depend in part on the extent to which the upwind footprint of the observations represents the actual conditions within the region encompassed by the model grid-cell, as well as on the extent to which the model input databases represent the vegetation characteristics within that grid cell. All chemistry and transport processes, including the forest mixing and shading examined here, thus, represent average conditions within the smallest volume resolvable by a model. Despite issues of spatial representativeness, transport of low concentration ozone from forested to non-forested regions will also impact total ozone levels

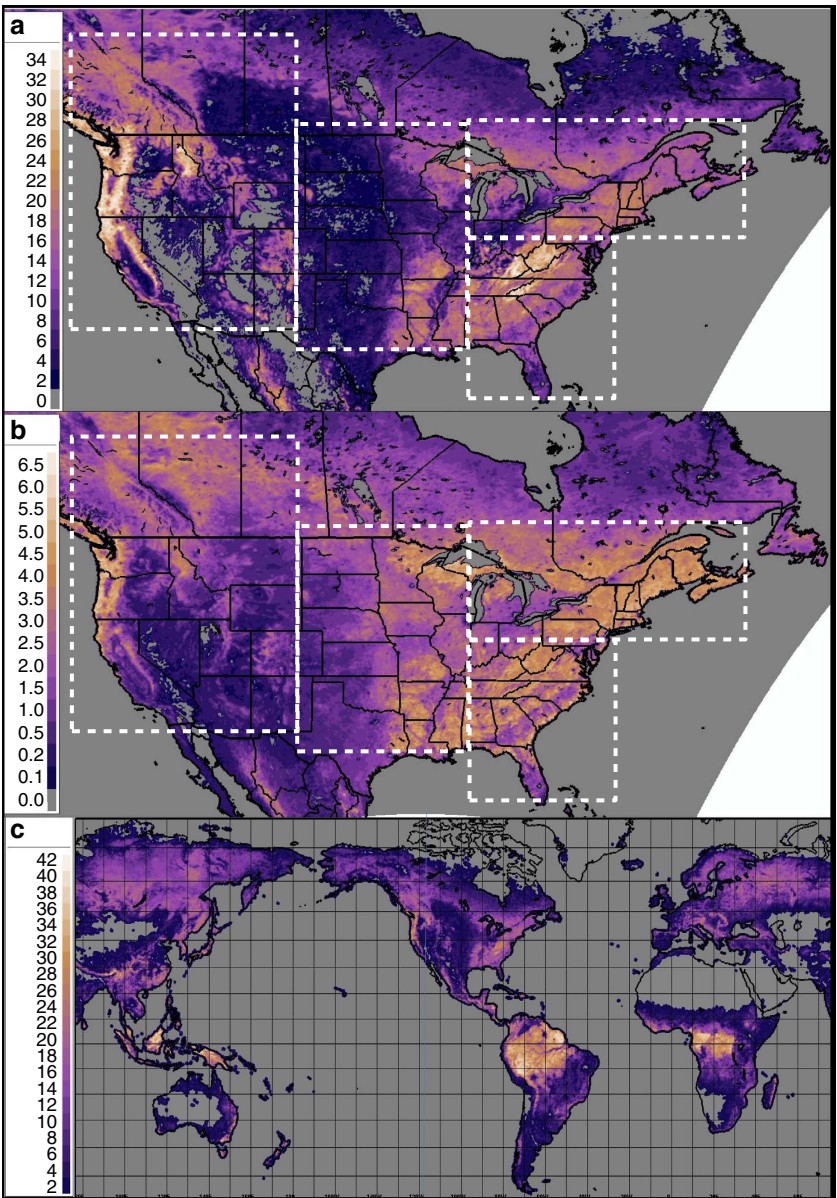

**Figure 3 | Satellite-retrieval-derived forest canopy data.** (**a**) Canopy height for North America (m)[37]; (**b**) Leaf area index for North America based on satellite retrievals (unitless); (**c**) Canopy height for the world (m)[37]. Dotted white lines indicate sub-region boundaries for Western, North America, Central USA, South-Eastern USA, and North-Eastern USA and Canada.

in the latter. With horizontal advection, low ozone concentrations in forested regions surrounding a city may thus reduce ozone levels in the latter.

**Forest canopy simulations using GEM-MACH.** We evaluated our hypothesis using our chemical transport model (GEM-MACH, versions 2 and 2.1), after modifying the model to include a canopy parameterization with two approaches for vertical diffusivity applied separately to the two model versions (see Methods). Four sets of simulations were carried out on GEM-MACH's operational 10 km resolution North American forecast domain; two 'base case' simulations using different versions of the unmodified model (versions 2 and 2.1, the latter including updates for urban heat island and the Obhukov Length, see Methods), and two simulations wherein vegetated canopy parameterizations were added to the base case simulations. The second of these canopy simulations investigated the extent to which changes to the shape of the expected profile of vertical diffusivity in response to stability might impact model performance. The simulations mimicked a three day forecast procedure, comprising consecutive, linked, three day forecasts at 0 and 12UTC spanning the month of July, 2010. Statistical evaluation of the model performance in each simulation was carried out using surface network hourly observations of ozone.

**The forest canopy and surface ozone.** The change in mean bias at observation stations between the canopy and base case simulations for GEM-MACHv2.1 is shown in Fig. 4a. Comparison with the corresponding 'base case' simulation of Fig. 1a shows that the forest canopy parameterization has significantly reduced the ozone bias in many of the regions with the highest positive bias in the original simulation, due to the prevalence of dense vegetation in those regions. The use of the forest parameterization decreases bias error to a level below that of the previous multi-model simulations (Fig. 4b), compare light grey (base case) versus dark grey (canopy) columns, for GEM-MACHv2 (base case: triangle, canopy: square) and GEM-MACHv2.1 (base case: circle, canopy: star). The normalized mean bias values in different regions of North America are also substantially reduced through the use of the canopy parameterization, in some regions to values very close to zero (Fig. 4c).

The change in the average ozone concentration at every model grid-cell is shown in Fig. 4d. The canopy parameterization has an impact throughout the model domain, usually resulting in ozone decreases, with the greatest decreases in the California and south-eastern USA forests, and lower decreases in the boreal forests of Canada. Ozone increases in Los Angeles, and slight increases may be seen in other areas. The impact of the canopy parameterization is widespread, and covers regions outside of the regions with high leaf area index (LAI) and canopy heights of Fig. 3a,b, such as the central plains of North America, and the downwind area of much of the Atlantic. This demonstrates that downwind transport of low-ozone air from canopy to non-canopy regions is sufficient to decrease ozone levels in the latter.

**The forest canopy and boundary layer ozone.** We have used ozone mass ratios from the third day of successive 0 UT simulations to show the influence of the canopy in the vertical dimension. The original hourly v2.1 model 3D ozone values were averaged on a daily and monthly basis, and (canopy/base case) ratios were constructed. A cross-section through these data, located in the centre of the most heavily forested part of eastern North America (Fig. 4b), is shown in Fig. 5 for the monthly mean ozone ratio (Fig. 5a), and the daily mean ozone ratio for July 4th

and July 27 (Fig. 5b,c, respectively). The figure shows that the influence of the forest canopy on ozone concentrations extends to heights far above the canopy layers or the lowest resolved original model layer, with ratios <1 throughout most of the atmospheric boundary layer.

Zones of lower ozone mass, due to the incorporation of the canopy parameterization, may be seen in all panels of Fig. 5, at both the lowest model layers and between heights of ∼850 to 690 mb (elevations of roughly 1,450–3,250 m above the surface). This overall depression of monthly average ozone mass in the upper part of the atmospheric boundary layer (Fig. 5a) is linked to shorter-term events in which lowest level air is transported upwards towards the top of the atmospheric boundary layer—examples of daily averages with these events may be seen in Fig. 5b,c. Roughly half of the daily averaged ratios constructed showed these events along this cross-section. Isolated high ratios in these cross-sections are linked to similar events occurring further upstream. Figure 5 shows that the canopy parameterization, due to this coupling between the lowest model layer and the rest of the atmospheric boundary layer, results in a monthly average ozone mass mixing ratio 'reduction' near the 'top' of the atmospheric boundary layer of up to 12%, and that daily average ozone mass mixing ratio reductions in the same area may reach 40%. These effects are in addition to the reductions near the surface which may be seen in Fig. 5, and are discussed and evaluated elsewhere in this work. An animation of the reduction in ozone resulting from the use of our forest canopy parameterization, with three-dimensional contouring of ozone concentration reduction factors of 0.95, 0.80 and 0.50, is included as Supplementary Movie 1.

This comparison shows that the forest canopy has a significant influence on ozone concentrations throughout the atmospheric boundary layer, due to interactions between the resolved scale meteorology and the region encompassed by the forest canopy itself.

The canopy processes examined here may help explain a known deficiency in global chemistry models employing data assimilation of satellite column ozone to attempt to improve model performance[27]. These efforts led to unwanted increases in positive ozone biases in the eastern USA, with the attribution of these effects to 'errors in the ozone sources or sinks in the boundary layer mixing in the model'. Our results suggest that at least part of these errors may be attributable to the forest canopy processes discussed herein, and that data assimilation efforts to improve tropospheric ozone forecasts would improve with the inclusion of forest canopy shading and turbulence, in global and regional air-quality models.

**Evaluation at Borden Forest Research Station.** The model's ability to capture the ozone profile at Borden Forest Research Station is shown in Fig. 6, in the lowest model layer resolved in the original vertical coordinate system (Fig. 6a,b; $R = 0.74$), and at 2 m altitude (Fig. 6c,d; $R = 0.59$). The decrease in ozone concentrations between the lowest layer resolved in the original coordinate system and 2 m is also shown (Fig. 6e,f, $R = 0.35$). While the main impact of the canopy parameterization is to reduce the ozone bias (as shown in Fig. 6a–d, in accord with the North American monitoring network evaluation shown above, the observed time series in the difference term (blue line, Fig. 6e) shows that differences between the layer average and 2 m ozone values have a significant time variation. The canopy parameterization captures at least some of the broad features of that variation (for example, compare blue to red line: high values in the difference between 2 and 9 July, increases in the difference between 12 and 14 July, 23 and 25 July and 27 and 29 July).

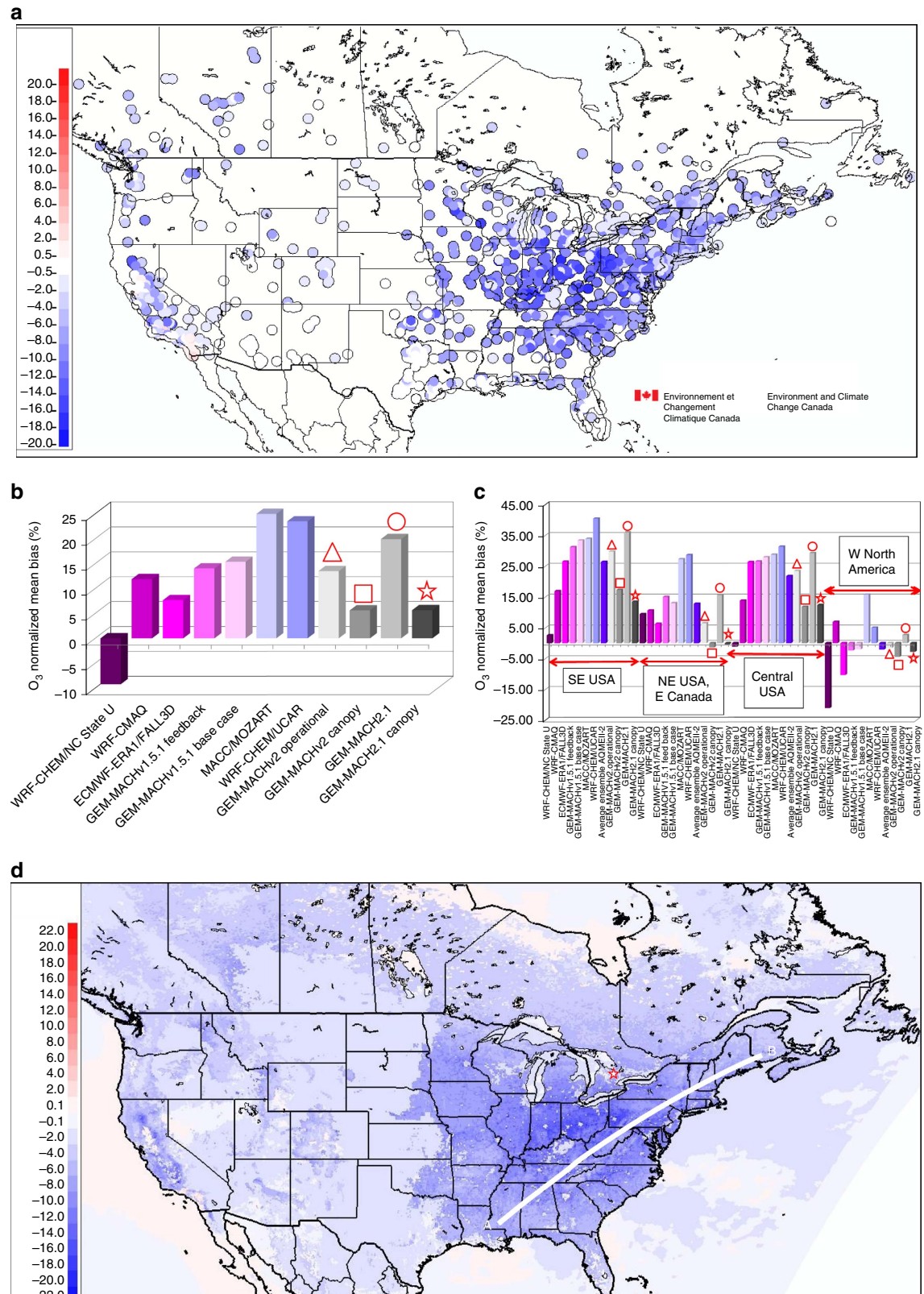

**Figure 4 | Effect of canopy shading and turbulence on ozone predictions. (a)** Difference in mean surface concentrations of ozone at observation stations (p.p.b.v.). **(b,c)** As in Fig. 1b,c, now including both base case and forest simulations from the current work. Simulations carried out as part of the current work are shown as light grey/dark grey column pairs overlaid with red symbols; triangle and square: GEM-MACHv2; circle and star: GEM-MACHv2.1. **(d)** Difference in mean surface concentrations of ozone at all model gridpoints (p.p.b.v.). The location of Borden Forest is shown by a red star in **d**, and the location of the Eastern North America vertical cross-section location described in the text is depicted as a white line (A–B). Both panels: (canopy parameterization—base case).

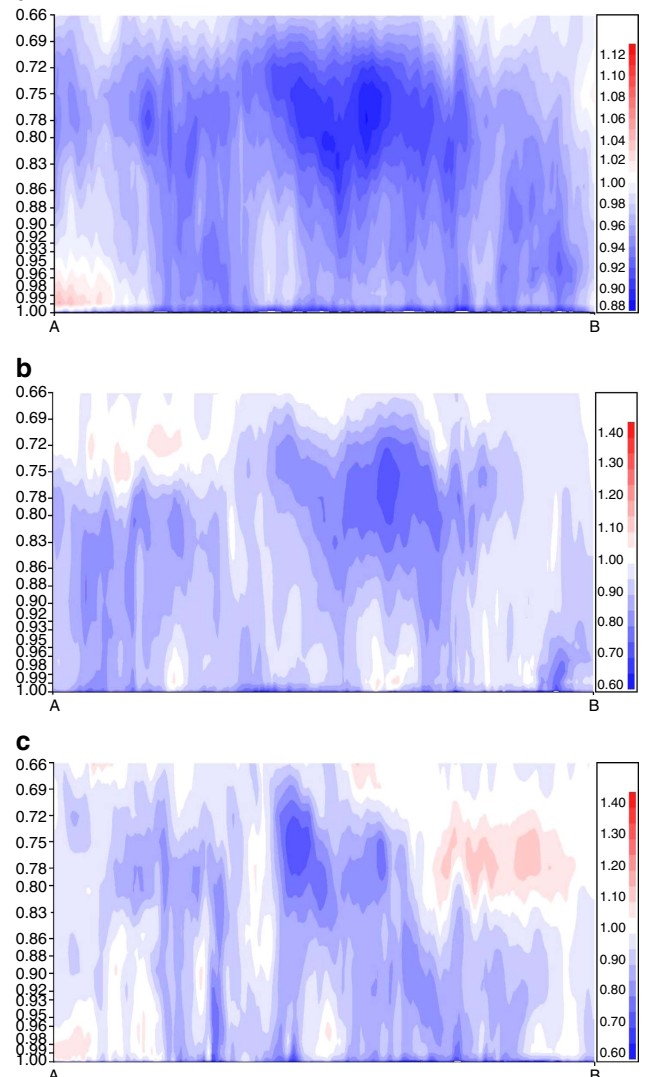

**Figure 5 | Canopy influence on atmospheric boundary layer ozone via canopy-to-base case mass ratio.** Eastern North-America cross-section of canopy-to-base ozone mass ratio, between the surface (hybrid level 1.0) and ∼3,450 m (hybrid level 0.66) (**a**) July 2010 monthly average. (**b**) Daily average for 4 July 2010. (**c**) Daily average for 27 July 2010. Vertical axis scale: hybrid coordinate (terrain following near the surface and pressure-following near the model top; a rough conversion to pressure in mb is to multiply the vertical axis values by 1,000).

The canopy parameterization shows at least some capability to capture the day-to-day variation in the daily differences which make up the average change in the mean bias, albeit at a relatively low correlation coefficient.

**Policy relevance of the forest canopy.** To place the magnitude of these changes in context relative to policy-relevant simulations from the literature, Fig. 7a compares the North American change in average ozone bias due to the forest canopy with the average ozone changes associated with other simulations designed to study feedbacks between weather and air-pollution[6] and European emissions reduction[28]. The vegetation-derived changes in ozone are of similar or larger magnitude than the changes described in the latter simulations. Figure 7b compares literature values for the largest local change in average ozone associated with the canopy effects to the largest local changes associated with

policy-relevant scenarios for climate change[29–31], and specific emissions reduction targets[32–34] both in North America and world-wide. The canopy parameterization has a larger local effect than these scenarios (−23 p.p.b.v., GEM-MACHv2), underscoring its importance for accurate simulation of the atmosphere, although we note that the magnitude of the changes from the referenced work may in part depend on the resolution of the underlying model framework. We note that these comparisons are for surface ozone changes only—while we have shown in Fig. 5 that the impacts of the forest on ozone levels extend throughout the atmospheric boundary layer, we are unable to show similar comparisons for the references quoted.

**Statistical evaluation of model performance.** A summary of the hourly ozone statistics[35] for North America for the four simulations are presented in Table 1. We find that the addition of forest canopy processes removes 59% of the positive bias in the operational GEM-MACHv2 ozone forecast, and 72% of the positive bias in the GEM-MACHv2.1 forecast. The two different canopy vertical diffusivity parameterizations also resulted in different levels of absolute decrease in the North American ozone mean bias; 2.50 p.p.b.v. for the v2 parameterization, and 4.51 p.p.b.v. for the v2.1 parameterization, suggesting that the shape of the within-canopy profile of the variance in vertical velocity in response to larger-scale atmospheric stability may have a significant impact on surface atmospheric chemistry. The forest canopy parameterization also improves the variance (reduced by one-third in the v2 simulations and by 8% in the v2.1 simulations). Some of the other statistics are also improved in the second to third decimal place (root mean square error, mean gross error, coefficient of efficiency and index of agreement). The model performance for the correlation coefficient, fraction of predictions within a factor of two, and covariance was degraded in the second decimal place. The largest relative change across the different error statistics was in the reduction in bias. As noted above and in Figs 1b,c and 4a, these improvements in bias performance are much higher when sub-regions dominated by forests are analysed. For example, the forest canopy processes reduce the magnitude of the bias of the GEM-MACHv2.1 in regions NA2, NA3 and NA4 by 58, 63 and 97%, while increasing the magnitude of the bias in region NA1 slightly (7%).

**Light attenuation versus canopy turbulence.** The relative importance of the reduction in photolysis rates versus resolved canopy turbulence was examined in a set of simulations in which only photolysis rates were modified. The gas-phase chemistry of the lowest resolved scale model layer was calculated twice at every time-step in the otherwise unmodified model, once using the original photolysis rates, and once using photolysis rates which had been attenuated with the full LAI depth of the forest canopy. The results of these two lowest layer calculations at each time-step were then averaged, effectively assuming instantaneous mixing between above- and below-canopy conditions. The results of this 'photolysis alone' simulation were compared to the base case, and resulted in a relative reduction of ozone mean bias of 19% (compared to 59% for the v2 simulation of Table 1). About one-third of the reduction in ozone concentrations may thus be attributed to the reduction in photolysis rates, and a further two-third of the reduction may be attributed to the separation in chemical regimes resulting from the turbulence parameterization and additional model layers.

We note that LES simulations of non-reactive tracers[25,26], coupled with our findings, have implications for the monitoring of ozone formation/destruction precursor chemicals: monitoring instrumentation located in clearings or small towns surrounded

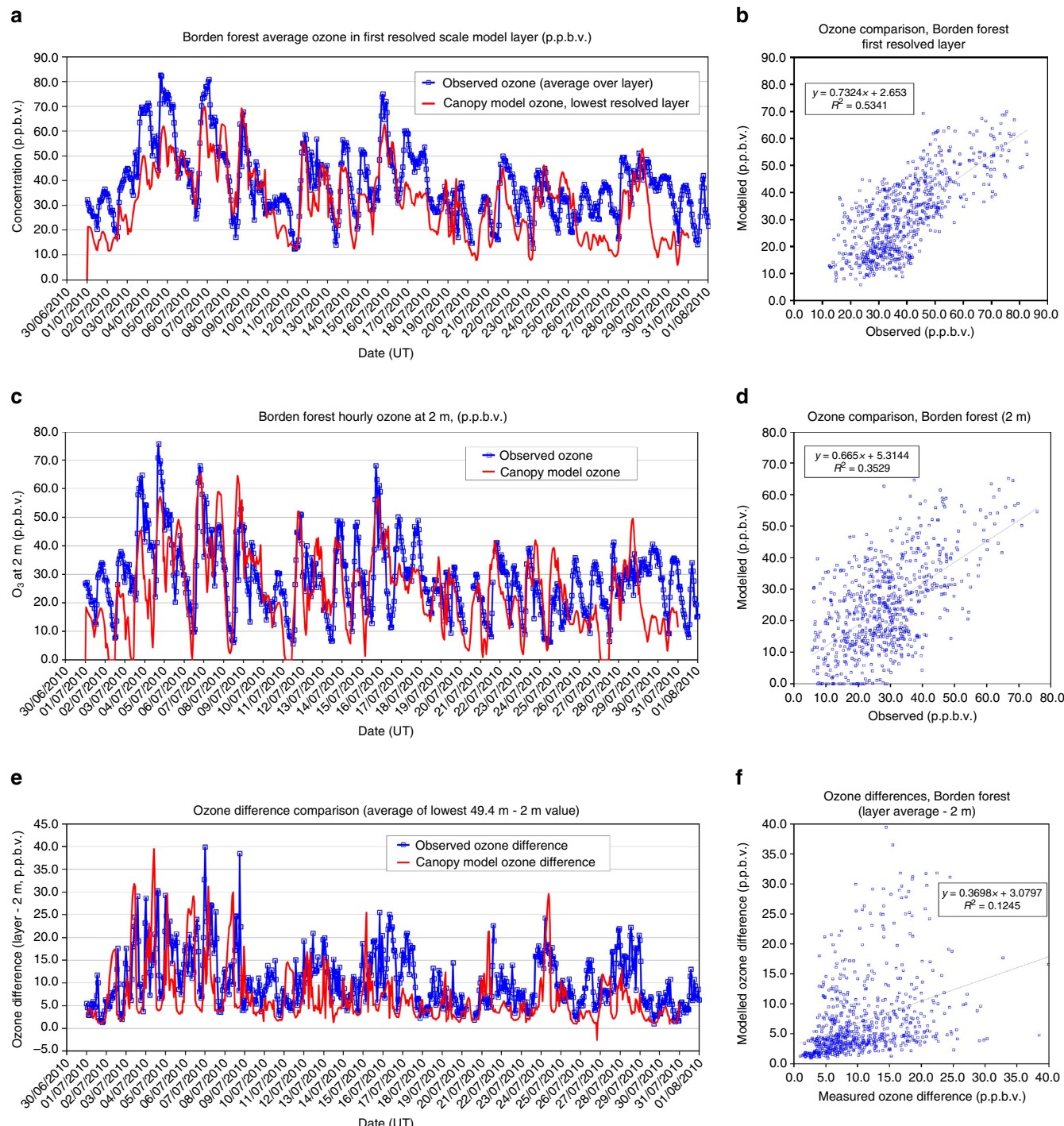

**Figure 6 | Borden Forest ozone evaluation.** (**a**,**b**) Time series and scatterplot comparing observed ozone over lowest model layer (0–49.4 m), compared to simulated values (p.p.b.v.). (**c**,**d**) Time series and scatterplot comparing observed and simulated values at 2 m (p.p.b.v.). (**e**,**f**) Time series and scatterplot comparing observed and simulated ozone decreases between original model resolution and 2 m values (p.p.b.v.).

by forest may report lower concentrations of these species than in the surrounding forests. Monitoring instrumentation at such locations may not be spatially representative of the surrounding terrain.

## Discussion
Our results are based on a single month's model simulation; longer simulations are planned. Further improvements to the parameterization may be possible with better forest classification, leaf-area and clumping index data. Our approach for

parameterizing within-canopy turbulence has been employed and evaluated in past one-dimensional models[36–39], but we note that the approach does not simulate some aspects of that turbulence (for example, counter-gradient fluxes[40]). Improvements to the manner in which within-canopy turbulence is parameterized within regional chemical transport models should thus be a focus for future work, along with measurement campaigns to improve on these parameterizations, and improvements to satellite retrieval algorithms used to recover the necessary model inputs for the parameterizations.

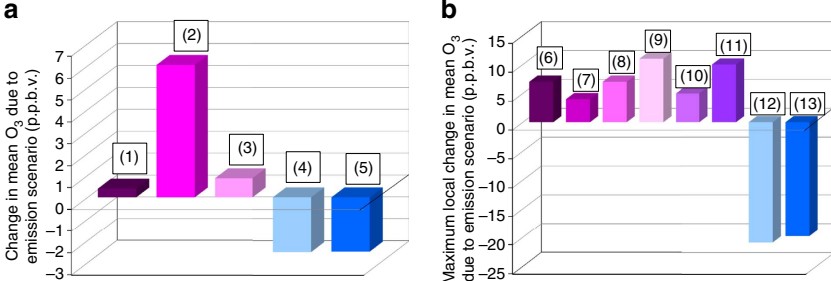

**Figure 7 | Comparison of changes in ozone associated with canopy simulations conducted here with other model predictions in the literature.** (**a**) Comparison of the changes in average ozone for feedback effects (direct and indirect) and emissions policy scenario simulations (purple shades) with the change in average ozone from the canopy simulations (blue). Columns (1)–(3), respectively: the change in average North American ozone due to feedback effects (direct + indirect)[6], the largest change in European 8 h average ozone for a specific emissions reduction scenario[28], and the smallest change in European 8 h average ozone for a specific emissions reduction scenario[28]. Columns (4) and (5), respectively: the change in mean North American surface ozone for the two forest canopy parameterizations examined here. (**b**) As in **a**, for change of local maxima in average ozone concentrations for literature scenario simulations (purple), compared to largest decrease in $O_3$ at North American monitoring stations, for the two canopy simulations (blue). Columns (6) through (11), respectively: the maximum change in the maximum daily 8 h average summer North American ozone change due to climate change, the maximum mean North American ozone change 2020s to 1990s, the maximum mean North American ozone change 2050 to 2000, the maximum mean summer world ozone change, 2030 versus 2000, the maximum change in local ozone associated with individual coal-fired power-plants, and the maximum change in local ozone due associated with a coal-fired power-plant in a single hour, respectively. Columns (12) and (13) correspond to the maximum change in station average ozone for the two canopy parameterizations considered here.

**Table 1 | Statistical analysis of model hourly ozone performance.**

| Performance metric | Simulation | | | |
|---|---|---|---|---|
| | GEM-MACHv2 | GEM-MACHv2 + canopy parameterization | GEM-MACHv2.1 | GEM-MACHv2.1 + canopy parameterization |
| Mean bias (p.p.b.v.) | 4.26 | 1.76 | 6.29 | 1.78 |
| Root mean square error | 15.13 | 14.64 | 15.81 | 14.79 |
| Mean gross error | 11.56 | 11.27 | 12.05 | 11.27 |
| Coefficient of efficiency | 0.19 | 0.21 | 0.15 | 0.21 |
| Index of agreement | 0.59 | 0.60 | 0.58 | 0.60 |
| Correlation coefficient (R) | 0.72 | 0.71 | 0.73 | 0.72 |
| Fraction of predictions within a factor of 2 | 0.79 | 0.78 | 0.80 | 0.78 |
| Standard deviation ($\sigma_M$) (observed $\sigma_O = 17.64$) | 20.66 | 20.12 | 20.75 | 20.62 |
| Variance | 9.12 | 6.15 | 9.67 | 8.88 |
| Covariance | 204.1 | 205.6 | 197.7 | 203.8 |

GEM-MACH, Global Environmental Multiscale-Modelling Air-quality and Chemistry.

Furthermore, we do not rule out other potential contributing factors to the ongoing positive ozone bias (such as errors in input emissions data). We have found in other work[41] that sub-grid-scale variability (the extent to which small towns and villages within forested regions are resolved by a given model resolution) can also significantly affect the canopy parameterization results; applying the parameterization to higher resolutions may also be beneficial. Our methodology should be treated as a first stage in what we hope will be an increased research focus on forest processes and their influence on regional and global chemistry.

Despite these limitations, our results show that the forest environment has a significant influence on atmospheric chemistry near the surface of the Earth, and throughout the atmospheric boundary layer, and are in accord with observations at forested sites. The resulting decreases in ozone have been shown to encompass a very large extent of North America, and will likely impact regions with significant forest canopies (from Fig. 3c, these include the northern boreal forests of North America and Russia, the world's rainforests and smaller regions, such as South-eastern Europe, South-west Asia and Japan).

## Methods

**Theoretical development.** First, the theoretical basis for the attenuation of light in vegetated canopies may be derived from Beer's Law as the probability of beam penetration (which may also be interpreted as the fractional light penetration) to a given level $z$ below the top of the canopy, using equation (1) below[21,22]:

$$P(\theta, z) = e^{-\frac{G(\theta)\,\Omega(\theta)\,\text{LAI}(z)}{\cos(\theta)}}, \quad (1)$$

where $P(\theta)$ is the probability of beam penetration at solar zenith angle $\theta$, $G(\theta)$ is the projection of unit leaf area in the $\theta$ direction (for a spherical leaf distribution, the usual assumption, $G = 0.5$), $\Omega(\theta)$ is the clumping index (a measure of the randomness of the leaf spatial distribution, $\Omega = 1$ if the leaves are randomly distributed), and $\text{LAI}(z)$ is the total LAI (the one-sided leaf area in the column per unit ground surface area) downwards from the canopy height to the given level within the canopy. Our approach is to use equation (1) as a proxy for the reduction in light associated with canopy shading; our photolysis rates being scaled down using the ratio of the above ($\text{LAI} = 0$) to within or below canopy values of equation (1). Note that we have made two approximations in the above—the diffuse component of the incoming radiation is assumed to attenuate in the same manner as the direct component, and the attenuation is assumed to be independent of wavelength.

Second, turbulence parameters such as the variance in the Eulerian vertical velocity, when plotted with a vertical coordinate scaled to the canopy height and scaled by parameters such as the friction velocity, show a remarkable similarity in profile shape across different vegetation types and canopy heights[23,40,42,43]. The localized near-field theory first proposed by Raupach[24] and adopted to one-dimensional canopy models[36–39] has been used here to approximate turbulent mixing throughout the canopy. We link the resulting diffusivity profiles to the resolved scale by their normalization to the above canopy diffusivity of the driving meteorological model. The approach and its limitations are discussed below (Coefficients of vertical diffusivity within the canopy).

Briefly, the overall approach taken followed four stages. In the first stage, satellite-retrieval derived LAI, $\Omega(\theta)$, canopy height data, vegetation fraction and population density data, were used to determine which model grid cells contain vegetated canopies (described in more detail below). The second stage comprised the use of vegetation-class-area weighted vertical profiles of LAI, and equation (1), to describe light attenuation for those model grid cells containing canopies, with this attenuation employed to reduce photolysis rates within and below the foliage. The third stage employed the above data to define the shape of profiles of the Eulerian vertical velocity throughout the canopy, which were in turn used to extend diffusivities resolved by the driving meteorological model down through the canopy. The final stage made use of additional model levels, added locally for canopy locations, to represent the within and below canopy region. This required splitting the core of the chemical transport model into two sets of model processes (one for all columns containing forest canopies, and one for all columns without forest canopies).

**Observations at Borden forest.** The Borden Forest Research Station is located in a mixed deciduous forest in Southern Ontario, Canada (44°19′N, 79°56′W, shown by a red star in Fig. 4d). The forest is a natural regrowth from farmland abandoned about 100 years ago. The forest consists of 52% red maple (Acer rubrum L.), 14% eastern white pine (Pinus strobes L.), 8% largetooth aspen (Populus grandidentata Michx.), 7% white ash (Fraxinus americana L.), 6% American beech (Fagus grandifolia) and 13% other species. The canopy height ($h$) near the flux tower is approximately 22 m with a peak LAI of $\sim 4.6\,\mathrm{m}^2\,\mathrm{m}^{-2}$ in summer. Long-term ozone measurements at the Borden Forest were conducted between 2008 and 2013 (refs 19,43).

Ozone profiles were measured by drawing air from 6 levels (41.5, 33.0, 25.7, 16.5, 5.3 and 1.0 m above ground) on a scaffold tower down 1.27 cm OD Dekabon tubes at a flow rate of $18\,\mathrm{l\,min}^{-1}$ to a switching manifold in a trailer at tower base. From there, $1\,\mathrm{l\,min}^{-1}$ was sub-sampled into trace gas analysers at 3 min intervals for each level. Ozone was quantified with a real-time ultravlet absorption instrument (Thermo Environmental Inc. Model 49C). The instrument was calibrated against NIST traceable standards on average every 6 weeks. Ozone losses in the sample line, quantified to be 9% through separate tests, were corrected to finalize the data.

PPFD above the canopy (33 m) was measured with a point quantum sensor (Li-190SA, Li-Cor Inc.), and below (1.5 m) the canopy with a line-integrating quantum sensor (Li-191, Li-Cor Inc.).

The value of $G \times \Omega$ in equation (1) at Borden was found to be 0.42; this is in reasonable agreement with the satellite-date derived value at the same model grid cell of 0.39.

**GEM-MACH simulations.** The details of this online chemical transport model's formulation may be found in the references provided above, along with a comparison of its performance relative to observations and peer-group models. As is the common practice with chemical transport models, specialized solvers are used for each component of the net differential equation describing a chemical's rate of change, combined using the operator splitting technique[44] (for example, gas-phase chemical rates of change are solved using a specialized 'stiff differential equation solver'). The model domain used here is a 0.09°- ($\sim 10\,\mathrm{km}$) resolution rotated latitude/longitude grid covering North America ($750 \times 620$ grid cells), used by Environment and Climate Change Canada for operational air-quality forecasts. The most recent operational configuration of the model (GEM-MACHv2) was used for our simulations. This version includes improvements to the modelling platform subsequent to the referenced material, notably improved mass conservation within the advection code, improved treatment of surface emission and deposition fluxes, and the addition of MOZART reanalysis-derived boundary conditions. Our second set of tests started from GEM-MACHv2.1 as a base case (this version of the driving model includes harmonization of Obhukov Length parameterizations between the driving meteorological and chemistry portions of the model, and a simple parameterization for the inclusion of anthropogenic heat sources on the model's radiative budget[45]). However, the only difference between 'canopy' and 'base case' simulations for our four simulations was the presence or absence respectively of the forest canopy parameterization within the model.

**Criteria for using a canopy parameterization.** Here we consider the conditions under which a vegetated canopy might be capable of altering the light and diffusivity conditions as noted above. Vegetated canopies which are too shallow to constitute a significant proportion of the first model layer (hence unlikely to influence the simulated chemical environment) can be ruled out, hence we exclude vegetated canopies less than 0.5 m in height. The canopy needs to be continuous in order to affect light and diffusivity significantly, hence we exclude grid cells for which the forest cover is less than ½ of the cell area, or for which more than 45% of the incident light for an overhead sun reaches the surface, and which have heights <18 m. We further assume that city core areas will not have vegetated canopies, hence regions with population densities greater than 50,000 per $10 \times 10\,\mathrm{km}$ grid cell are eliminated. Very low LAI values imply little impact on shading or diffusivity—hence the lower limit of LAI for which the parameterization is applied is 0.1. The grid cells not excluded by these criteria are treated using the canopy parameterization.

**Canopy layer structure and vegetation vertical distribution.** For vegetated canopies, we add three additional layers to the vertical structure of the model, intended to represent the main changes in canopy vertical diffusivity and LAI in the vertical dimension (the latter influencing equation (1) at each layer), at $z = h_c$, $z = 0.5 h_c$ and $z = 0.2 h_c$ where $h_c$ is canopy height for the given grid cell, obtained from satellite retrievals[46]. The parameterization requires the vertical distribution of leaves within the canopy in order to determine the photolysis rates for each successive canopy layer. Typical leaf vertical distributions from the literature[47–50] for deciduous, coniferous and mixed forests were assigned to the 230 Biogenic Emissions Landuse Database, version 3 (BELD3) vegetation classifications[51]; land-use fraction weighted combinations of these values were thus used to create grid-cell-specific LAI vertical distributions ( see Supplementary Data 1 and 2). Similarly, land-use-weighted values of clumping index were derived by assigning initial estimates from the literature[52–57] to the BELD3 land-use classifications, these values were adjusted to create BELD3 clumping index values matching MODIS satellite retrieval estimates[55] (Supplementary Data 1). Spatially gridded vertically integrated total LAI was derived from MODIS retrievals[58], as described below.

The resulting LAI distributions, clumping index values were applied using equation (1) to determine the attenuation of light at each canopy layer interface and midpoint, hence deriving vertically integrated net attenuation weights for each layer. These weights were subsequently applied to the above-canopy photolysis rates, creating attenuated photolysis rates for each canopy layer. Attenuation was also applied to PPFD using the BEIS3 emissions algorithms[2]; downward cumulative LAI values and the LAI distributions were used to determine the PPFD penetrating to each layer, and the mass of biogenic emissions from each layer. These were emitted directly into the given layers rather than as a surface-based mass flux, aside from NO emissions, which are derived from soil sources, and remained as a surface flux.

**Coefficients of vertical diffusivity within the canopy.** Our procedure for assigning turbulence within the forest canopy follows the near-field theory first proposed by Raupach[24] and adopted to one-dimensional canopy models[34,36,38,39], scaled to diffusivities provided by the driving meteorological model. The approach makes use of the following formulae[24] to describe the canopy vertical diffusivity profile shape:

$$K_{\mathrm{can}}(z) = \frac{K_{\mathrm{mod}}(z_1)}{K_{\mathrm{est}}\left(\frac{z_1}{h_c}\right)} K_{\mathrm{est}}\left(\frac{z}{h_c}\right), \qquad (2)$$

$$K_{\mathrm{est}}\left(\frac{z}{h_c}\right) = \sigma_{\mathrm{w}}^2\left(\frac{z}{h_c}\right) T_{\mathrm{L}}\left(\frac{z}{h_c}\right), \qquad (3)$$

$$T_{\mathrm{L}}\left(\frac{z}{h_c}\right) = \frac{h_c}{u^*}\left[0.256\left(\frac{z - 0.75 h_c}{h_c}\right) + 0.492 \exp\left(\frac{-0.256 z / h_c}{0.492}\right)\right], \qquad (4)$$

In the above equations, $z$ is the height above the Earth's surface, $z_1$ is the height of the lowest model layer before the application of the canopy parameterization, $h_c$ is the canopy height, $u^*$ is the friction velocity, $T_{\mathrm{L}}$ is the Lagrangian timescale and $K$ is the vertical diffusivity. The variance in the Eulerian vertical velocity ($\sigma_{\mathrm{w}}^2$) was defined in two ways; for GEM-MACHv2 simulations, a profile chosen to represent a real canopy under neutral conditions was used[24]

$$\sigma_{\mathrm{w}}\left(\frac{z}{h_c}\right) = \begin{cases} 1.25 u^* & , \frac{z}{h_c} > 1.0 \\ u^*\left[0.75 + 0.5*\cos\left(\pi\left(1 - \frac{z}{h_c}\right)\right)\right] & , \frac{z}{h_c} \le 1.0 \end{cases}, \qquad (5)$$

For GEM-MACH2.1 simulations, we noted that a better fit to observations than (5) would be to combine the average profile of $\sigma_{\mathrm{w}}\left(\frac{z}{h_c}\right)$ from multiple observation studies[23] with the observation that the shape of the $\sigma_{\mathrm{w}}$ profile with height 'flattens' with increasing stability[59]. Stability is controlled at scales larger than the canopy by the driving meteorological model, which in turn sets the magnitude of the diffusivities via the ratio expressed in equation (2). The key concern with equations (5), and (6) through (9) which follow, is that they adequately represent the shape of the resulting profile in $\sigma_{\mathrm{w}}$, hence the profile of diffusivity, which in turn is normalized by the meteorological model's diffusivity at the resolved scale.

Our approach was to follow the available observations[59] of $\sigma_{\mathrm{w}}$ profiles to the greatest extent possible. First, we noted that the $\sigma_{\mathrm{w}}/u^*$ profile of equation (5) (ref. 24) when over-plotted with the data from multiple neutral conditions observation studies[23,42], tended to over-predict the magnitude of $\sigma_{\mathrm{w}}$ below the canopy height. The coefficients of (5) were adjusted to create a $\sigma_{\mathrm{w}}/u^*$ profile centred in the distributions of these observation data[23,42]. The resulting equation for this observation-centred $\sigma_{\mathrm{w}}/u^*$ profile was

$$(-0.1 \le h_c/L < +0.1):$$

$$\sigma_{\mathrm{w}}\left(\frac{z}{h_c}\right) = \begin{cases} 1.0 u^* & , \frac{z}{h_c} > 1.25 \\ u^*\left[0.625 + 0.375*\cos\left(\frac{\pi}{1.06818}\left(1.25 - \frac{z}{h_c}\right)\right)\right] & , 0.175 \le \frac{z}{h_c} \le 1.25 \\ 0.25 u^* & , \frac{z}{h_c} \le 0.175 \end{cases},$$

$$(6)$$

For the GEM-MACHv2.1 simulations, equation (6) was used for neutral atmospheres ($-0.1 \leq h_c/L < +0.1$), given that the corresponding suite of observations were for neutral atmospheric conditions.

Observations of $\sigma_w/U(34\,\text{m})$ were segregated into neutral and stable environments were available from one measurement study[59]. These profiles showed that above the canopy, the rate of increase in $\sigma_w/U(34\,\text{m})$ with decreasing stability (that is, going from stable to neutral conditions) was much larger above the canopy than near the surface. Ratios of the values of $\sigma_w(\frac{z}{h_c}=1.8)/\sigma_w(\frac{z}{h_c}=0.3)$ for stable and neutral conditions were constructed from these data, resulting in ratio values of 4 and 2.5 for stable and neutral conditions, respectively. The neutral atmosphere value of this ratio from (6) is 4.0, conforming to these observations. For stable atmospheres, we noted that the trend of the flattening of the $\sigma_w$ with increasing stability[59] would imply a flat profile (no variation in $\sigma_w$ with height) for stability levels with $h_c/L > 0.9$. We therefore set the slope of $\sigma_w$ versus $u^*$ to be constant for very stable conditions:

$$h_c/L \geq 0.9 :$$
$$\sigma_w = 0.25u^*, \quad (7)$$

In order to have a smoothly varying change in the profile shape between neutral and very stable atmospheres, equation (6) was modified, so that the coefficients determining its vertical variation would asymptotically approach the neutral atmosphere solution at $h_c/L = 0.1$, and the very stable atmosphere solution at $h_c/L = 0.9$:

$$0.1 \leq h_c/L < 0.9 :$$
$$\sigma_w\left(\frac{z}{h_c}\right) = \begin{cases} 0.25\left(4.375 - 3.75\frac{h_c}{L}\right)u^* & , & \frac{z}{h_c} > 1.25 \\ u^*\left[A + B^* \cos\left(\frac{\pi}{1.06818}\left(1.25 - \frac{z}{h_c}\right)\right)\right] & , & 0.175 \leq \frac{z}{h_c} \leq 1.25 \\ 0.25u^* & , & \frac{z}{h_c} \leq 0.175 \\ \text{where} \\ A = 0.125R + 0.125 \\ B = 0.125R - 0.125 \\ R = 4.375 - 3.75\frac{h_c}{L} \end{cases}, \quad (8)$$

We note that this formula provides an above-canopy to surface level $\sigma_w$ ratio of 0.5 when $h_c/L = 0.5$, in accord with the available observations[59].

The available data[59] categorized the change in $\sigma_w$ for stable (defined in that reference as $h_c/L > 0.5$) and neutral ($-0.1 \leq h_c/L < +1$) environments, but no data were available for unstable environments. The observed change in the above-canopy to surface ratio in $\sigma_w$ from stable to neutral conditions implies that the ratio may increase further with further decreases in stability. Rather than extrapolate far beyond the available observations, we chose a ratio of above-canopy to surface $\sigma_w$ of 5.0 for unstable conditions, constructing the final of the four profiles using this assumption:

$$h_c/L < -0.1 :$$
$$\sigma_w\left(\frac{z}{h_c}\right) = \begin{cases} 1.25u^* & , & \frac{z}{h_c} > 1.25 \\ u^*\left[0.75 + 0.5*\cos\left(\frac{\pi}{1.06818}\left(1.25 - \frac{z}{h_c}\right)\right)\right] & , & 0.175 \leq \frac{z}{h_c} \leq 1.25 \\ 0.25u^* & , & \frac{z}{h_c} \leq 0.175 \end{cases}, \quad (9)$$

This last assumption may result in an underestimate of the 'trapping' of pollutants in unstable environments if the relative ratio in above-canopy to surface $\sigma_w$ values is > 5.0. Additional observations under unstable conditions are needed to improve this version of our parameterization further. Also, the intent of the profiles is to create the shape of the profile of diffusivity with the assumption that the diffusivity at the resolution of the driving meteorological model is correct. From the spread of observed $\sigma_w$ values[23,42], further refinements based on more specific types of vegetation are possible.

For both implementations of $\sigma_w(\frac{z}{h_c})$, the values of friction velocity from the driving meteorological model, the canopy height, and equations (3) through (5) (GEM-MACHv2) or equations (3,4) and (6–9) (GEM-MACHv2.1) are used to create an estimate of the vertical diffusivity at the driving meteorological model's first layer above the ground ($K_{\text{est}}(\frac{z_1}{h_c})$). The ratio of the driving meteorological model's vertical diffusivity at this height ($K_{\text{mod}}(z_1)$) to the estimate is used to scale vertical diffusivities downward through the canopy from height $z_1$, using equations (2) through (9). In both implementations, the equations thus ensure that the scaled 'shape' of the canopy profile is in accord with observations across multiple types of vegetated canopies[23], and, for GEM-MACHv2.1, attempt to account for canopy-specific changes in atmospheric stability[59]. The subsequent scaling of the resulting profile to meet the diffusivity of the driving meteorological model at height $z_1$ thus allows an approximation to the diffusivity within the canopy based on the available observations of $\sigma_w$, while taking into account differences associated with changes in canopy height, friction velocity and the response of canopy turbulence to atmospheric stability.

As noted above, the meteorological model's predicted stability influences its predicted diffusivity coefficients. These diffusivity coefficients are used to normalize the derived diffusivity profile within the parameterized canopy layers via equation (2). The impact of large-scale stability is thus incorporated into the parameterized diffusivities within the canopy. The second variation on the

parameterization, described in equations (6) through (9), attempts to account for the additional influence of stability on the profile of the variance in Eulerian vertical velocity within the canopy itself[55]. For this second variation, we note that few if any observations of $\sigma_w$ aggregated for highly unstable conditions exist; our 'unstable' profile is thus similar to the neutral profile, with the bulk of the changes associated with unstable conditions being the result of the resolved scale meteorology's diffusivity values. Further observational evidence is needed to improve the method for highly unstable conditions.

It should be noted here that our two approaches for estimating $\sigma_w$ are not a result of Monin–Obukhov theory. Numerous authors[60–64] have shown that Monin–Obukhov theory does not adequately represent canopy turbulence. Instead, we use fits to observed $\sigma_w$ profiles and the relationship[24] $K = \sigma_w^2 T_L$ to infer the shape of the profile, which is then normalized to allow a smooth transition to resolved model layer $K$ values above the canopy.

Other approaches have been put forward for estimating canopy transport in the context of equilibrium solutions for the velocity profile, specifically above an inferred displacement height[65], with a review of these methods appearing in the literature[66]. These approaches compare well to observations above the inferred displacement height, although they are not applicable to the region within the canopy below that displacement height. Nevertheless, they represent another avenue for future improvements to canopy processes. The dependence of canopy turbulence on LAI has also been examined in the literature[67]. When sufficient observation information on specific canopies is available to allow characterization of this methodology's free parameters, this approach has been shown to provide a good fit to observed canopy turbulence, although performance is degraded in cases of insufficient observation accuracy and for complex canopies. Nevertheless, we feel that this approach and the incorporation of LAI into canopy turbulence parameterizations should be pursued in future work, in conjunction with the collection of additional measurement data for parameterization evaluation.

Efforts to describe forest turbulence processes have also taken place making use of LES models, where the effects of the forest on the momentum equation are included as a leaf area density-dependent drag term, for very high-resolution domains. Grid cells sizes typically on the order of meters, and very small computational time-steps, prohibit the use of these models as parameterizations within a regional chemical transport model, but they are useful tools for generating potential parameterizations. For example, modifications[68] to an existing LES model[69], where drag imposed by a forest canopy was added to the momentum balance equation, was used to simulate multiple layers within the plant canopy. While the resulting model compared well to past Borden measurements[70] no attempt was made to create a parameterization of the results in a manner suitable for incorporation into regional chemical transport models, unlike past LES efforts for the convective boundary layer[71]. More recently, a LES model for contiguous forest canopies[72] was used to demonstrate the ability of such models to simulate coherent turbulent structures similar to those seen in observational studies (though no direct comparisons to observations were made). The potential for significant edge effects along forest edges was also demonstrated[26] using a LES model, with simulated flow features similar to those seen from observations, although no formal evaluation against observations was carried out. LES models to date have not been used to create forest mixing parameterizations which would be suitable for the coarse time and spatial resolutions of regional chemical transport models. However, they do have the capability to simulate counter-gradient diffusion within forest canopies, an observed process which cannot be simulated with the current down-gradient diffusion approach used here. A potential future direction for this work is thus the use of LES models to create improved parameterizations for canopy turbulence at regional chemical transport model resolution.

**Splitting of the model core.** The subroutines handling GEM-MACH's gas, aqueous and particle chemistry, vertical diffusion, emission, deposition and particle microphysics are solved sequentially using operator splitting[44] on vertical slices within horizontal tiles of the model domain (the tiles being distributed across groups of processors using the message passing interface (MPI) paradigm, and for individual slices within each tile using the OpenMP paradigm). Once canopy versus no-canopy grid cells are identified and the additional vertical layers added to the canopy-containing model columns, all of the subsequent model integration processes aside from advection were also applied independently to 'canopy' and 'no canopy' columns for each model time step, for each of the model operators within GEM-MACH's model core.

One disadvantage of the canopy parameterization is that it introduces an additional set of model layers which are spatially discontinuous in the horizontal dimension. The advection processes of the model, however, require a continuous, regular grid. For this reason, advection is carried out on the original model layers. At the end of the each time step of the chemistry processes outlined above, the relative contributions of the mass of each tracer towards the original model layer mass are stored during the process of reassigning model layer mass to original model layers. These model ratios are used to redistribute the mass from the advected model layers back to the canopy layers at the start of the subsequent time step. This is an approximation necessitated by the nature of the advection solver, but also justified by observations and theoretical development[23] which show that vertical transport within the canopy is dominated by turbulence rather than advection. The vertical distribution of mass within canopy layers will thus be

controlled by chemistry and vertical diffusion, and will not be significantly modified relative to these terms by advection.

**Conversion of satellite data to gridded model input.** The parameterization makes use of satellite retrievals of canopy height and LAI as input parameters. The canopy height data[46] were originally available as 1 km resolution pixels; these were aggregated to the 10 km resolution of GEM-MACH by calculating the average of the 1 km resolution canopy heights enclosed by each model 10 km resolution grid cell. Total vertically integrated LAI values were derived from the MODerate Resolution Imaging Spectrometers (MODIS) on the NASA Terra and Aqua satellites[58,73]. These archived MODIS LAI values were derived from surface reflectance observations in the red and near-infrared combined with other input information and look-up tables calculated using radiative transfer models. Here, we used archived 8-day MODIS Terra/Aqua composites for July 2010 at 1 km resolution (product code MCD15A2, Collection Five[74]). The 1-km resolution satellite pixel values spanning July 2010 were aggregated to monthly totals on the GEM-MACH 10 km continental grid.

**Statistical metrics used for evaluation.** This section describes the statistical metrics used in model evaluation[35] appearing in Table 1. In the following, $M_i$ is the model value, and $O_i$ is the observed value, for station 'i'.

Fraction of predictions within a factor of 2: the relative fraction of model values that fall within a factor of two of the observed value (1.0 for a perfect model).

$$0.5 \leq \frac{M_i}{O_i} \leq 2.0, \tag{10}$$

*Mean Bias.* This is the average of the difference (model—observation) for all data pairs; negative numbers indicate that the model values on average are lower than observations, positive values indicate that the model values on average are higher than observations (0.0 for a perfect model). Mean bias values are in the units of the observed variable.

$$MB = \frac{1}{N} \sum_{i=1}^{N} M_i - O_i, \tag{11}$$

*Mean Gross Error (also known as the Mean Absolute Error).* The average magnitude of the difference between model and observations (0.0 for a perfect model). Mean gross error values are in the units of the observed variable.

$$MGE = \frac{1}{N} \sum_{i=1}^{N} |M_i - O_i|, \tag{12}$$

*Root Mean Square Error.* The s.d. of the differences between the model and observed values. The RMSE is a measure of model accuracy (0.0 for a perfect model, units same as the variable being evaluated).

$$RMSE = \sqrt{\left( \frac{\sum_{i=1}^{N} (M_i - O_i)^2}{N} \right)}, \tag{13}$$

*Pearson Correlation Coefficient.* A measure of the degree of linear dependence between two variables; a perfect linear relationship between observations and model values will have a positive slope and a correlation coefficient of $+1.0$. In the following formula, $\sigma_M$ and $\sigma_O$ are the model and observed standard deviations, respectively.

$$r = \frac{1}{(N-1)} \sum_{i=1}^{N} \left( \frac{M_i - \bar{M}}{\sigma_M} \right) \left( \frac{O_i - \bar{O}}{\sigma_O} \right), \tag{14}$$

*Coefficient of Efficiency.* This measure allows a comparison of the magnitude of the difference between model and observed values relative to the magnitude of the difference between the observed values and the observed mean. A perfect model will have a score of 1.0, while a score of 0.0 indicates that the mean of the observed data is as accurate a predictor as the model values (no time-dependent predictive advantage of the model).

$$COE = 1.0 - \frac{\sum_{i=1}^{N} |M_i - O_i|}{\sum_{i=1}^{N} |O_i - \bar{O}|}, \tag{15}$$

*Index of Agreement.* This measure compares the magnitude of the difference between model and observed values relative to the difference between the observations and the observed mean (that is, the sum of the error magnitudes relative to the deviation of the observations from their mean). A perfect model would have a score of $+1.0$.

$$IOA = \begin{cases} 1.0 - \frac{\sum_{i=1}^{N} |M_i - O_i|}{2 \sum_{i=1}^{N} |O_i - \bar{O}|}, & \text{when } \sum_{i=1}^{N} |M_i - O_i| \leq 2 \sum_{i=1}^{N} |O_i - \bar{O}| \\ \frac{2 \sum_{i=1}^{N} |O_i - \bar{O}|}{\sum_{i=1}^{N} |M_i - O_i|} - 1.0, & \text{when } \sum_{i=1}^{N} |M_i - O_i| > 2 \sum_{i=1}^{N} |O_i - \bar{O}| \end{cases}, \tag{16}$$

*Variance.* The square of the difference in the spread of the variables from their mean value[12]:

$$VAR = (\sigma_M - \sigma_O)^2, \tag{17}$$

*Covariance.* The degree of correlation between the modelled and observed time series[12]:

$$COV = 2(1-R)\sigma_M \sigma_O, \tag{18}$$

**Data availability.** The data that support the findings of this study are available from the corresponding author upon request.

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

## Acknowledgements

We acknowledge the work of colleagues M.D. Moran, S. Gravel and V. Savic-Jovcic, for leading the operational implementation of version 2 of GEM-MACH.

## Author contributions

P.A.M.: project lead, study hypothesis and underlying theoretical development, code modifications, carrying out model simulations, model evaluation and drafts of manuscript. R.M.S.: Borden observation data set lead, generation of averages used in Fig. 2 and observations used in Fig. 6. A.A.: code modifications and debugging, carrying out model simulations, statistical analysis and graphical analysis. J.Z.: emissions generation, LAI data filtering and GIS merging of satellite data with model grids. C.M.: satellite data retrieval. S.K.K.: satellite data retrieval and work-up. B.P.: model simulations. P.C.: model simulations. Q.Z.: GIS merging of satellite data with model grids.

## Additional information

**Competing interests:** The authors declare no competing financial interests.

