## [Peer Review File · Nature Communications]

Reviewers' Comments:

Reviewer #1 (Remarks to the Author)

This paper addresses the issue of how the forest canopy impacts near surface ambient ozone levels. This has been an active area of research for many years, with many studies focused on biogenic emissions and photochemistry related issues. This paper points out that current efforts to predict near surface ozone show high positive bias, attributed to many factors ..deposition, emissions, etc.

This paper focuses on 2 processes impacted by the canopy .. reduced photolysis and reduced turbulence. The paper describes in detail how these processes are added to the model, and through case study simulations , show that including these processes improved the predictions and significantly reduced the bias.

The paper is well written and analyzed. I feel that the paper is written for the atmospheric chemistry community and will motivate the modeling community to add in these processes into other models and to refine/improve the parameterizations and ability to constrain the parameters needed to tune the parameterizations.

It is an excellent piece of work, but I feel it perhaps is best suited for a more specialized journal (say GRL) .. based on this being a case study .. model based study

Reviewer #2 (Remarks to the Author)

See attached pdf.

Reviewer #3 (Remarks to the Author)

Re: Review of "Darkness Under the Trees: The Forest Canopy and Surface Ozone" by Dr Makar and colleague.

Air quality modelling, especially at regional scale, suffers an historic lack of visibility although it is intrinsically multidisciplinary encompassing physical, chemical and mathematical modelling along with emission estimates, land use retrieval and meteorology. The paper reviewed here might help reaching out the broader geophysical modelling community.

The paper is overall neat and well written, and clearly fits the scopes of the journal. I found a few aspects that, in my opinion, require further attention, and that I invite the authors to consider in order to make the results more robust and their exposure clearer.

Once the necessary changes have been made and questions addressed, I see no objection for accepting the manuscript for publication in Nature Communications.

General remarks

1. The large (sometime dominant) influence of emission uncertainty, anthropogenic and natural NO_x, as well as isoprene is not properly highlighted, in my view. The authors should make an effort to explain why the high bias they observe cannot be explained by the error in the emissions. For example, there is high bias in the 'Central USA' region (figure 1a), although figures 2d and 2e show that the region is not forested. On the same line, literature results show that RMSE error for modelled ozone is similar in Europe and North America, although the forest impact in Europe might be much smaller, according to Figure 2f. In my view, while in Europe that of the emission's error is a big issue, it might not be the case in North America, but it cannot be excluded from the discussion as it stands. I invite the authors to comment on this point.

2.The influence of forest canopy on ground-based monitoring stations is not clear to me, since the stations are not sited within the forest. Within the canopy the vertical mixing is inhibited and concentration much lower than aloft, but how that information is carried to the monitoring station,

outside the forested area, is not clear. Somehow, the authors should explain that monitoring stations falling within model grid cells interested by the canopy parameterization have a spatial representativeness that includes the effect of the forest (if that is the case...), and that volume-averaged values (that of the models) are compared against point measurements (that of the station). Unless I have misunderstood, this point I believe is very important. Otherwise the model bias could, again, be explained by biased-high emissions in the forested areas. Line 91 indeed says '...limits the vertical mixing of chemicals emitted near the earth's surface.' But (anthropogenic) pollutants are not emitted within the forest, but rather in the grid cell containing a fraction of forest, right? Hope I have explained my doubt.

3. Merging the previous two comments, I invite the authors to show in figure 3 only the results at stations falling within model grid cells interested by the canopy parameterization. The other stations should not report any change, right? Or maybe add contour lines for the 'zero'.

Specific comments (in no particular order)

- I believe it should be said clearer that conclusions are entirely drawn based on one month model simulation.
- Any effort made to simplify the exposure of the method and results would be beneficial. For example I would invite the authors to consider changing the naming of the modelled cases to 'base' and 'base+canopy', and to produce results for one model version only. I can't see the advantage of having version 2 and version 2.1, since the focus of the study is the canopy parameterization. I believe this doubling of the results can bring confusions and distract from the main goal, especially considering that the two versions produce similar statistical scores.
- Line 111. '...has significantly reduced the ozone bias in the forested regions...'
- Line 138-139. The correlation coefficient not changing is somehow expected, as the new canopy module only acts on the bias, and the correlation is bias-independent. However, it shows that the new parameterization does not feed the model new covariance error, in light of the bias-variance trade-off theorem (Krogh, A., Vedelsby, 1995). Just a comment.
- Although not fully justified, the use of parametric statistics is customary in air quality model evaluation. I would encourage the authors to use the following three metrics: squared bias (or mean bias) for distance, squared ($\text{var}(\text{obs}) - \text{var}(\text{mod})$) for variance, and $(1-r)\sigma_{\text{mod}}\sigma_{\text{obs}}$ for covariance (σ is the standard deviation and var the squared standard deviation). The summation of the three returns the total MSE. All other (parametric) metrics are derived from these three (see Murphy et al 1988; Gupta et al 2009) and are redundant.
- Eq (1). I am no expert, but I believe that this equation would later need an ad-hoc evaluation.
- Item 2, Line 171-177. I agree with the statement here. Please consider reading Finnigan (2000); Cionco (1965).
- Line 232. Were the urban areas detected by using the modeled heat flux? How reliable is that? Would not be feasible to use imagery analysis?
- Line 258. What about the lateral exchange at the edge of the forest (in light of main comment #2)?
- Statistical scores. Please specify the time aggregation implied in the spatial summation over the stations.

Some editorial suggestions

Line 57. Please consider ',...measurements of ozone. concentration in the forest....differences at different heights...'

Line 59. Please provide reference to you statement ending with 'foliage'.

Line 60. Please provide reference to you statement ending with 'environments'.

Line 61. 'O3'. For consistency should be 'ozone'.

Line 73. What do the authors mean by 'light levels'? Is it the intensity, the squared amplitude, please specify.

Line 88. '...relatively to non-vegetated...?'

Line 89. Please consider '...surfaces, the attenuation of downward radiation due to foliage, combined with inhibited vertical diffusivity, suppresses...'

Line 99. The references for GEM-MACH have been already provided at line 41, although different ones. Please verify.

Line 100-104. This I find confusing. According to me it should read: '...two base-case simulations to which the vegetated canopy parameterization was added, for a total of four sets of model simulations...', or something on that line.

Figure 3a. 'Shading' has not been defined/introduced before. Maybe better keep 'canopy'.

Eq(1) - LAI should read LAI(z), according to the detail given later, am I right?

Line 179. '...to determined which model grid cells contains forest canopy'. I would add ' and what fraction'.

Line 209. See previous 'Line 99' comment.

References

- Finnigan, 2000. Turbulence in plant canopies. *Annual Review of Fluid Mechanics*, Vol. 32: 519-571
- Cionco R (1965) A mathematical model for air flow in a vegetative canopy. *J Appl Meteorol* 4: 517-522
- Gupta, V.H., and et al., 2009. Decomposition of the mean square error and NSE performance criteria: implications for improving hidrological modelling. *Journal of Hidrology* 377, 80-91
- Murphy, A., 1988. Skill scores based on the mean square error and their relationships to the correlation coefficient. *Monthly Weather Review* 116, 2417-2424
- Krogh, A., Vedelsby, J.: Neural network ensembles, cross validation, and active learning, in: *Advances in Neural Information Processing Systems*, 7, 231-238, 1995.

Our detailed responses to the Reviewers' comments follows.

Reviewer #1:

This paper addresses the issue of how the forest canopy impacts near surface ambient ozone levels. This has been an active area of research for many years, with many studies focused on biogenic emissions and photochemistry related issues. This paper points out that current efforts to predict near surface ozone show high positive bias, attributed to many factors ..deposition, emissions, etc.

This paper focuses on 2 processes impacted by the canopy .. reduced photolysis and reduced turbulence. The paper describes in detail how these processes are added to the model, and through case study simulations , show that including these processes improved the predictions and significantly reduced the bias.

The paper is well written and analyzed. I feel that the paper is written for the atmospheric chemistry community and will motivate the modeling community to add in these processes into other models and to refine/improve the parameterizations and ability to constrain the parameters needed to tune the parameterizations.

It is an excellent piece of work, but I feel it perhaps is best suited for a more specialized journal (say GRL) .. based on this being a case study .. model based study

We thank the reviewer for the kind comments. With regards to the specialization level of the journal to which we have submitted the article: the work identifies a key large-scale atmospheric process which to date has been studied only on the local scale via observation studies at specific sites, and sometimes with 1 dimensional canopy models (we have contributed to these local studies in the past; see the references for Makar et al JGR 1999, Froelich et al, 2015, Wu et al, 2016, in the revised article). The impact of the processes on the large-scale chemistry of the atmosphere has gone unrecognized until now. In order to create the parameterization used here, new forms of data based on satellite retrievals were required. The work thus links three separate scientific communities – the regional/global modellers, forest emissions/turbulence/chemistry observations community, and the satellite data community. The work suggests new avenues of research for all of these communities – e.g., improving regional AQ models through better parameterizations (we regard our work as a first try, and certainly hope others will build on the idea), carrying out surface measurement studies with the specific aim of devising those improved parameterizations, and devising new means of using space-based information for the same aim of improving understanding of forest environment interactions with large-scale atmospheric chemistry. The co-authors on our paper come from each of these communities – but we recognized that to reach all of them, we would need to submit to a journal with a broader interest base. Hence our decision to submit the work to the Nature family of journals. We’ve added text to this effect in the summary paragraph of the revised manuscript, in the main body of the paper: “Our results are based on a single month’s model simulation; longer simulations are planned. Further improvements to the parameterization may be possible with better forest classification, leaf-area and clumping index data. Our approach for parameterizing within-canopy turbulence has been employed and evaluated in past one-dimensional models (Makar et al,1999; Stroud et al,2005; Gordon et al, 2014), but we note that the approach does not simulate some aspects of that turbulence (e.g. counter-gradient fluxes (c.f. Finnigan, 2000)). Improvements to the manner in which within-canopy turbulence is parameterized within regional chemical transport models should thus be a focus for future work, along with measurement campaigns to improve on these parameterizations, and improvements to satellite retrieval algorithms used to recover the necessary model inputs for the parameterizations. Furthermore, we do not rule out other potential contributing factors to the ongoing positive ozone bias (such as errors in input emissions data). We have found in other work (Makar et al, 2016) that sub-grid-scale variability (the extent to which small towns and villages within forested regions are resolved by a given model resolution) – can also significantly affect the canopy parameterization results; applying the parameterization to higher resolutions may also be beneficial. Our methodology should be treated as a first stage in what we hope will be an increased research focus on forest processes and their influence on regional and global chemistry.

Despite these limitations, our results show that the forest environment has a significant influence on atmospheric chemistry near the surface of the earth, and are in accord with observations at forested sites. The resulting decreases in ozone have been shown to encompass a very large extent of North America, and will likely impact regions with significant forest canopies (from Figure 3(c), these include the northern boreal forests of North America and Russia, the world's rainforests, and smaller regions such as south-eastern Europe, south-west Asia, and Japan).

”

Reviewer #2:

This manuscript discusses the influence of a canopy parameterization on model predicted near-surface ozone concentration biases. The topic is timely and is certainly of interest for the community. However, I have some serious reservations about the findings described in the manuscript:

- *While I fully appreciate the attempt to introduce vertically-varying canopy-induced processes on the turbulent eddy diffusivity, the authors never provide any evidence that their method reproduces their observed within-canopy reactant profiles. So, I am unable to ascertain whether the bias improvements shown are physically meaningful or simply a tuning exercise.*

The reason why this was not included into our original manuscript was due to the transient nature of the canopy levels within our canopy parameterization: as noted on original manuscript page 17, line 309, the canopy levels only exist within computer memory for the duration of each timestep, prior to their values being returned to the model's original vertical structure. However, the predicted ozone values at 2m elevation above the ground and the average model concentrations within the first 49.4 m are retained, and were used in our revised manuscript to evaluate the parameterization's ability to recover the vertical structure of ozone within the canopy. Bearing in mind that this is a comparison at a single location as opposed to the surface values for the entire continent already evaluated, we have added a new figure to the manuscript (new Figure 5), showing the model performance at Borden Forest for the month of July. We compare the observed and simulated ozone predicted by the model for the lowest 49.4 m of the atmosphere (Figure 5(a,b)), the values observed and predicted at 2m (Figure 5(c,d)), and the difference between the lowest 49.4m and the 2m ozone values, for both observations and predictions (Figure 5(e,f)). The following text has been added to the manuscript with respect to this figure:

“The model's ability to capture the ozone profile at Borden Forest Research Station is shown in Figure 5, in the lowest model layer resolved in the original vertical coordinate system (Figure 5(a,b); $R = 0.74$), and at 2m altitude (Figure 5 (c,d); $R=0.59$). The decrease in ozone concentrations between the lowest layer resolved in the original coordinate system and 2m is also shown (Figure 5(e,f), $R=0.35$). While the scatter about the line of best fit is the largest in Figure 5(e,f), the canopy model shows a significant capability to simulate the decrease in ozone compared to the original model (where the lowest layer is assumed homogeneous, that is, the difference is always zero).”

- *It has also been shown that organized structures dominate the exchange of scalars between the canopy-layers and aloft (e.g., Gao et al. 1989, Raupach et al. 1996, Finnigan et al. 2009). Therefore, non-local transport generally contributes significantly to within-canopy diffusion. Holtslag and Moeng (1991) put forward a strategy to incorporate non-local transport within turbulent eddy diffusivity parameterizations. Given the importance of non-local transport in the canopy and since this currently presented parameterization neglects non-local transport, I again find myself uncertain if the improvements shown O_3 biases arise for physically meaningful reasons.*

The reviewer brings up an important branch of forest turbulence research which we should have referenced in the original manuscript, and have referenced in the current manuscript, that of Large Eddy Simulations (LES) of forest canopy processes. We agree that non-local transport may be an important factor, but we are constrained by the spatial and temporal scales of our model domain to use relatively simple parameterizations. Parameterizations of counter-gradient diffusion specifically designed for forest canopies do not seem to be available in the literature; the LES community has yet to provide a parameterization which would be suitable for forest canopies within the (relatively) coarser resolutions used in regional chemical transport modelling. We do acknowledge that this issue should be discussed in the manuscript, since the use of LES models may provide a means for creating such a parameterization.

With regards to the reviewer's reference, Holtslag and Moeng (1991) used an LES model (a 5x5x2km domain model with 96 gridpoints in each dimension (Moeng and Wyngaard, 1989)) to derive a parameterization for countergradient turbulence which gave reasonable agreement to observations of $\overline{w^2\theta}$ and $\overline{w\theta^2}$ within the convective boundary layer. The vertical resolution of the model was thus 48m, and, importantly from our viewpoint, no attempt was made to account for canopy processes within this LES – the resulting recommended non-local transport parameterization is thus inappropriate for use with forest canopies.

However, other references describe forest-specific LES simulation papers. We have included the following text within the revised manuscript to acknowledge these efforts: “Efforts to describe forest turbulence processes have also taken place making use of Large Eddy Simulation (LES) models, where the effects of the forest on the momentum equation are included as a leaf area density dependent drag term, for very high resolution domains. Grid cells sizes typically on the order of meters and very small computational time-steps, prohibit the use of these models as parameterizations within a regional chemical model, but they are useful tools for generating potential parameterizations. For example, modifications⁶³ to an existing LES model⁶⁴, where drag imposed by a forest canopy was added to the momentum balance equation, was used to simulate multiple layers within the plant canopy. While the resulting model compared well to past Borden measurements⁶⁵ no attempt was made to create a parameterization of the results in a manner suitable for incorporation into regional chemical transport models, unlike past LES efforts for the convective boundary layer⁶⁶. More recently, a LES model for contiguous forest canopies⁶⁷ was used to demonstrate the ability of such models to simulate coherent turbulent structures similar to those seen in observational studies (though no direct comparisons to observations were made). The potential for significant edge effects along forest edges was also demonstrated⁶⁸ using a LES model, with simulated flow features similar to those seen from observations, though no formal evaluation against observations was carried out. LES models to date have not been used to create forest mixing parameterizations which would be suitable for the coarse time and spatial resolutions of regional chemical transport models. However, they do have the capability to simulate counter-gradient diffusion within forest canopies, an observed process which cannot be simulated with the current down-gradient diffusion approach used here. A potential future direction for this work is thus the use of LES models to create improved parameterizations for canopy turbulence at regional chemical transport model resolution.”

- *The authors discuss Monin-Obukhov Similarity Theory (M-O) and its use to ascertain the canopy-top friction velocity – a critical parameter in their parameterization. However, it has been clearly demonstrated over the past decades that M-O underpredicts the relationship between a scalar gradient and the flux of that scalar over tall canopies (e.g., Garratt 1978, Raupach 1979,*

Denmead and Bradley 1985, 1987, Cellier and Brunet 1992, among numerous others). Numerous strategies have been put forward over the years to account for the canopy-induced processes thought to be responsible for having modified the relationship between fluxes and gradients in the vicinity of tall canopies (see for example: Raupach 1994, M^older et al. 1999, Harman and Finnigan 2007, 2008, Nakai et al. 2008, De Ridder 2010). Weligepolage et al. (2012) evaluated a number of the parameterizations using observed data and found that Harman and Finnigan (2007) most faithfully reproduced their observations (likely resulting from their having incorporated a new length scale associated with the canopy-induced organized turbulent motions). So, I wonder – with all these options available to the current authors – how they justify relying on straight M-O to represent surface exchange above their tall plant canopies and how their necessarily under-predicted scalar fluxes impact the O₃ biases presented.

We do not use straight M-O theory as suggested by the reviewer, since we are aware that M-O theory would not be able to predict the shape of the σ_w profile within the canopy. Rather, our approach follows the localized near-field theory first proposed by Raupach (1989) and adopted to our 1D canopy models in Makar et al (1999), Stroud et al (2005) and Gordon et al (2014). We've attempted to clarify our methodology in the revised text to make this point clear. Recall that our methodology uses fits to observed σ_w profiles to generate a representative diffusivity profile. We then normalize the diffusivity profile to the meteorological model profiles for K at the first resolved model layer, in order to achieve a smooth transition in diffusivity between the canopy and the region above. Our reason for this approach is two-fold:

- (1) To use σ_w observations as the basis for our approach to the greatest extent possible, hence avoiding issues such as the need to correct M-O theory for canopy conditions, and
- (2) The requirement that our diffusivities above the canopy are consistent with the driving model meteorology (to avoid the creation of spurious discontinuities in diffusivity).

With regards to the second requirement: most chemical transport models do not explicitly include meteorological calculations but instead make use of meteorological data output from a pre-existing meteorological simulation. A key issue for our parameterization is thus to be able to approximate canopy mixing from the output parameters from an existing meteorological simulation.

The (bulk) friction velocity provided by the driving meteorological model was thus used for the canopy-top u^* in the parameterization. We note that Simpson et al (1998) compared u^* values at multiple measurements heights of 46, 33 and 25 m above the Borden canopy height of 20m, and found relatively little variation in u^* with height above the canopy (46 to 33m: slope of 1.06, intercept -0.06, R^2 0.93; 33 to 25m: slope of 1.01, intercept 0.02, R^2 0.95). Both of the approaches we took to describe the shape of the subsequently normalized vertical diffusivity profile are based on best fits to observed σ_w profiles within and above forest canopies. In the second of the two approaches, the observed “flattening” of the σ_w profile with increasing stability, shown Shaw et al (1988), Figure 16, was used to approximate the stability effect on the shape of the σ_w profile. This last reference chose to categorize the profiles of σ_w according to a stability definition using the ratio of canopy height to observed Obhukov length. The “dependence” of our methodology on M-O theory is thus at most indirect (the dynamics in our driving weather forecast model GEM employ M-O to create the Obhukov Length and to relate the wind profile to u^*). The relative importance of the manner in which stability is incorporated can be seen by comparing our two approaches. The first approach used the same profile (neutral conditions) regardless of model-predicted stability (or how stability was characterized), while the second approach included the flattening of the σ_w profile shape to follow Shaw et al (1988)'s observations. Both approaches resulted in very similar ozone performance (Table S13 in the original manuscript, Table 1 in the revised manuscript), though the

performance relative to the two respective base cases implies that the second parameterization accounted for a greater net reduction in mean bias than the first parameterization. From this we infer that the additional stability correction on the σ_w profile shape had a net positive influence on the model results, though we also note in the revised manuscript that additional observations of σ_w under unstable conditions are required to improve the parameterization further.

We investigated the approach of Harman and Finnigan (2007) as recommended by the reviewer: our hope was to determine whether that approach could also be easily used or modified to generate a diffusivity profile for extension downwards within the canopy via σ_w . However, we found that the shape of the resulting diffusivity profile can't reproduce the observations of Patton et al (2003) throughout the below-canopy region, hence we have identified it and similar methods for future follow-up, though we have not used them here. The difficulty with Harman and Finnigan (2007) and similar methods is that they are designed to describe the flow field in the upper part of the canopy, to an inferred displacement height, but not below that height. We present our analysis of Harman and Finnigan (2007) below (all numbered equations are with reference to Haman and Finnigan (2007)):

- (1) The constant β was assumed to be 0.3, this being a reasonable value for u^*/U_h (equation 5).
- (2) The value of H was assumed to be 22 m, and the value of the LAI was assumed to be 4.0. Using the authors' recommendation that $L_c = 4 H / \text{LAI}$, the values of $l = 2\beta^3 L_c$ and $d_t = \beta^2 L_c$ are 1.188 and 1.98 respectively, and $L_c = H = 22\text{m}$.
- (3) We assume values of $L = -100\text{m}$ and $L = +100\text{m}$ for testing purposes (note that L_c / L thus lies within the authors' range noted as falling within observations at Duke forest).
- (4) We use the authors' functions for correcting the M-O part of the solution (ϕ_m) with the roughness sub-layer correction function $\widehat{\phi}_m$:

$$\Phi_m(z) = \phi_m\left(\frac{z+d_t}{L}\right) \widehat{\phi}_m\left(\frac{z+d_t}{l}\right) \quad (11)$$

We note here that Harman and Finnigan (2007) use a shifted vertical z coordinate with zero at the canopy height, hence the surface is at $-H$, where H is the height of the canopy. However, the formulae should thus only be valid for $z > -d_t$, though Harman and Finnigan (2007) show figures for heights $z/L_c = -1$; i.e. down to the surface level, in our case. See for example, H&F(2007) Figure 2b, which shows the profile of $U(z)/u^*$. It is unclear to us how equation (11) can be used to generate $U(z)$ below $z = -d_t$. Perhaps the reviewer could help clarify this apparent contradiction. We note that Weligepolage et al. (2012), using the parameterization of Harman and Finnigan (2007) in a comparison of similar approaches – none of which were used to provide $U(z)$ profiles extending to the ground. A key requirement for us is to be able to define the values of the diffusivity throughout the canopy and down to the surface.

Given that the L value is negative, we use the authors' recommended formulae for $\phi_m\left(\frac{z+d_t}{L}\right)$:

$$\begin{aligned} \phi_m\left(\frac{z+d_t}{L}\right) &= \left[1 - 16\left(\frac{z+d_t}{L}\right)\right]^{-\frac{1}{4}}, L < 0 \\ \phi_m\left(\frac{z+d_t}{L}\right) &= \left[1 + 5\left(\frac{z+d_t}{L}\right)\right], L \geq 0 \end{aligned} \quad (10)$$

Following the authors' derivation, we have

$$\widehat{\phi}_m\left(\frac{z+d_t}{l}\right) = 1 - c_1 \exp\left[-\beta c_2 \left(\frac{z+d_t}{l}\right)\right] \quad (19)$$

And that

$$\Phi_m(0) = \frac{\kappa}{2\beta} \quad (16)$$

Which in turn leads to their equation for c_1 (top of their page 352, no equation number):

$$c_1 = \left\{ 1 - \frac{\kappa}{2\beta\phi_m(z=0)} \right\} \exp\left[\frac{c_2}{2}\right]$$

The constant c_2 is set to $\frac{1}{2}$ following the authors' recommendation, and the above equation may be solved to find the value of c_1 for the stability function $\phi_m(z=0)$. For example, for the unstable case examined here ($L=-100$):

$$c_1 = \left\{ 1 - \frac{\kappa}{2\beta} \left[1 - 16 \left(\frac{d_t}{L} \right)^{\frac{1}{4}} \right] \right\} \exp\left[\frac{1}{4}\right]$$

For the above values of β , L (-100), and d_t , we have $c_1 = (1 - 0.2/0.3 * 1.071223) * 1.2840 = 0.36703$.

Equation (11) in this test case with $L = -100$ thus becomes:

$$\Phi_m(z) = [1 + 0.16(z + 1.98)]^{-\frac{1}{4}} \{1 - 0.36703 \exp[-0.1263(z + 1.98)]\} \quad (A)$$

For the case $L=+100$, the alternate $\phi_m(z=0)$ is used:

$$c_1 = \left\{ 1 - \frac{\kappa}{2\beta} \left[1 + 5 \left(\frac{d_t}{L} \right)^{-1} \right] \right\} \exp\left[\frac{1}{4}\right] = 0.50512 \quad (B)$$

Hence

$$\Phi_m(z) = [1 + 0.05(z + 1.98)] \{1 - 0.50512 \exp[-0.1263(z + 1.98)]\} \quad (C)$$

Equations (A) and (C) can be used to define eddy viscosities in the surface layer, viz:

$$K_m(z) = \frac{\kappa(z+d_t)u^*}{\Phi_m(z)} \quad (D)$$

Taking $U_h = 10 \text{ ms}^{-1}$, $u^* = 3 \text{ ms}^{-1}$, and the formulae for $K_m(z)$ become:

$$K_m(z) = \frac{1.2(z+d_t)}{[1+0.16(z+1.98)]^{-\frac{1}{4}} \{1-0.36703 \exp[-0.1263(z+1.98)]\}} \quad \text{for } L=-100, \text{ and} \quad (E)$$

$$K_m(z) = \frac{1.2(z+d_t)}{[1+0.05(z+1.98)] \{1-0.50512 \exp[-0.1263(z+1.98)]\}} \quad \text{for } L = +100 \quad (F)$$

Our original thought was to use the above two formulae for $K_m(z)$ to determine the shape of the diffusivity profile, for comparison to the Patton et al values of σ_w in our approach (given that we make use of Raupach's $K = \sigma_w^2 T_L$, a K profile generated from Harman and Finnigan (2007), normalized to the value at the canopy height should have a similar shape). However, on inspection, it can be seen that all of the above formulae are only applicable for heights above $z = -d_t$; they are undefined or produce negative values throughout much of the canopy (z from $-d_t$ to $-H$). Plots of the K values from the above formula for $z > -d_t$ are shown below. Note that neither plot shows the observed and expected behavior of the diffusivity given the σ_w observations (relatively little change with height close to the ground, a sharp gradient within the foliage, with increases thereafter dominated by the T_L term):

We also investigated Weligepolage et al. (2012), who incorporated Harman and Finnigan (2007)'s approach into their intercomparison with wind (U) observations for a single forested test case. In that review of similar methods, it can clearly be seen that all of the methods compared fail to capture the within-canopy wind velocities: all of the approaches examined, including Harman and Finnigan (2007), were shown to predict zero wind speeds 18m above the ground (see left-hand figure below). The observations put together by Patton et al (2003) (see right-hand figure below) show that the actual winds decrease more gradually with decreasing height. The parameterizations investigated by Weligepolage et al (2012), including that of Harman and Finnigan (2007), do not provide accurate information within the forested canopy, and are only valid for the region above the height $z=-d_t$.

Figure R2.2. Comparison of Weligepolage et al (2012)

Figure 5(a), Weligepolage et al, (2012). Calculated wind speeds using different approaches; solid line = Harman and Finnigan (2007). Note that U approaches zero at 20 m (canopy height = 32m). Second scales added for comparison to graph at right.

Figure 4(a) from Patton et al (2003): U/U_h where U_h is the wind speed at the canopy height, for different observation studies. Note that in most cases, U/U_h is non-zero even close to the surface.

Given the above investigation, we don't think it appropriate to incorporate Harman and Finnigan (2007) into our parameterization; their methodology does not extend throughout the canopy column, as is required for our model simulations.

We do, however mention both the M-O issue and the Harman and Finnigan (2007) approach and references, along with their recommendation for more detailed investigation, in the revised paper, with the following text. First, the start of the description of the σ_w terms now reads:

"Our procedure for assigning turbulence within the forest canopy follows the near-field theory first proposed by Raupach (1989) and adopted to 1D canopy models (Makar et al, 1999; Stroud et al, 2005; Gordon et al (2014), scaled to diffusivities provided by the driving meteorological model. The

approach makes use of the following formulae²⁴ to describe the canopy vertical diffusivity profile shape:”

We then continue towards the end of the section:

“It should be noted here that our two approaches for estimating σ_w are not a result of Monin-Obukhov theory. Numerous authors (cf. Garratt 1978, Raupach 1979, Denmead and Bradley 1985, 1987, Cellier and Brunet 1992) have shown that Monin-Obukhov theory does not adequately represent canopy turbulence. Instead, we use fits to observed σ_w profiles and the relationship $K = \sigma_w^2 T_L$ (Raupach, 1989) to infer the shape of the profile, which is then normalized to allow a smooth transition to resolved model layer K values above the canopy.

Other approaches have been put forward for estimating canopy transport in the context of equilibrium solutions for the velocity profile specifically above an inferred displacement height (Harman and Finnigan (2007), with a review of similar methods in Weligepolage *et al.* (2012)). These approaches compare well to observations above the inferred displacement height, though they are not applicable to the region within the canopy below that displacement height. Nevertheless, they represent another avenue for future improvements to canopy processes.”

Others in the literature (e.g., Hamba 1993, Vil`a-Guerau de Arellano et al. 1995, Patton et al. 2001, among numerous others) have clearly shown that the presence of the chemical production/loss terms in the conservation equation for reacting species ensures that the relationship between a reactant’s vertical concentration gradient and its vertical flux must differ from that for non-reacting scalars (which do not contain a reactivity term). Therefore, the turbulent eddy diffusivity must necessarily vary with the effective reaction rate for each species. For slowly reacting species, the turbulent eddy diffusivity shouldn’t differ terribly much from that for non-reacting scalars, but it should differ substantially for more rapidly reacting species many of which are emitted by vegetation (e.g., isoprene, monoterpenes, etc) and which react with OH to alter O3 production/destruction rates. However, the parameterization discussed here presumes that similarity for all scalars, reactive or not. The disconnect between the canopy-induced photolysis rate modifications and the turbulent eddy diffusivity in this parameterization therefore seems terribly lacking.

We agree with the author on the importance of chemical losses within the forest canopy and of the importance of their inclusion in all models, and in the estimation of emission fluxes of reactive hydrocarbons, having demonstrated their importance ourselves in previous work 17 years ago (Makar et al, “Chemical processing of biogenic hydrocarbons within and above a temperate deciduous forest, J. Geophys. Res., 104, D3, 3581-3603, 1999) and two subsequent papers referenced in the revised manuscript. However, we disagree with the approach to deal with the issue suggested by the reviewer for two reasons:

- (1) The suggested approach does not follow the commonly accepted numerical approach of “operator splitting” used by all chemical transport models (operator splitting allows the inclusion of reactive gases without the need to create diffusivities which vary with reactivity), and
- (2) The creation of diffusivities that vary with chemical reactivity does not conform to commonly accepted physics on the diffusive transport of gases.

With regards to the first point, the commonly accepted practice in creating models of atmospheric chemistry and transport (described in textbooks on atmospheric modelling such as Jacobson, M.Z., Fundamentals of Atmospheric Modeling (2005) and Seinfeld, J.H., and Pandis, S.N., Atmospheric Chemistry and Physics, 2016), is to solve the different processes contributing to the rate of change of a given chemical in sequence, the outcome of each process being used as an initial concentration

for the subsequent process. The seminal work on this topic was carried out by Marchuk, G.I., *Methods of Numerical Mathematics* (1975, 1982), wherein the error magnitude of this approximation was shown to be a function of the order of the operators and the size of the time step employed. Operator splitting allows the use of highly specialized solvers for the different parts of the net differential equation (e.g. gas-phase chemistry and turbulent transport may be solved with stiff differential equation solvers and a relatively simple one-step backward difference solution employing a tridiagonal matrix solver, respectively). Operator splitting is also necessary for large scale computation, since solving the full coupled system of equations would require matrix inversion operations on the order of the number of gridpoints multiplied by the number of chemical species – which is far too large for efficient computation. Consequently, all reaction-transport models make use of operator splitting to carry out numerical prediction of atmospheric chemistry, following Marchuk, 1982. GEM-MACH, like all other chemical transport models, solves the system of differential equations for gas-phase chemistry within a stiff differential equation solver, and then solves the vertical diffusion equation for each species (using the same coefficients of vertical diffusivity across all species) in a subsequent part of the model code. The impact of gas reactivity is thus included in a formal, mathematically proven and evaluated fashion, following the common practice within the chemical transport modelling community. As we showed in our earlier work (Makar et al, 1999) for the case of a 1D reactive transport model, which also makes use of operator splitting, this allows one to accurately simulate the impact of reactivity on individual gases within a forest canopy, and determine the relative impact of diffusive mixing versus chemical transformation on canopy chemical species.

With regards to the second point: when the reactive effects are incorporated into the net solution of the system of differential equations, there is no need for parameterized reactivity-dependent diffusivities.

So, while we agree with the reviewer's comment that chemical reactivity is important, we disagree with the reviewer's assertion that we have not included reactivity effects. We have done so, by making use of specialized chemical solvers to carry out that part of the solution of the net system of equations, and the accepted practice of operator splitting. In order to make the work more accessible to the broader community, we have included a short description of operator splitting citing the above references into the revised manuscript, viz:

"As is the common practice with chemical transport models, specialized solvers are used for each component of the net differential equation describing a chemical's rate of change, combined using the operator splitting technique (Marchuk, 1982) (e.g. gas-phase chemical rates of change are solved using a specialized "stiff differential equation solver").

I find it quite unsettling that there doesn't seem to be much discussion throughout the manuscript regarding the parameterization and its derivation/application to unstable stratification regimes, even though it is during unstable daytime periods when photochemical processes are most important and which the manuscript proposes to solve.

The reviewer's assertion that photochemical processes during unstable daytime periods are the most important is incorrect. The parameterization impacts daytime periods via both the reduction in photolysis rates and the revised profile in diffusivity mixing and increased resolution in the canopy. The parameterization *also* influences the nighttime chemistry, due to the revised profile in turbulence. The interaction between chemistry and turbulence in this case is as follows. The relevant chemical reactions are:

Nitrogen monoxide (NO) is emitted by combustion sources close to the surface of the earth. This reacts with radicals such as RO₂ to create nitrogen dioxide (reaction 1); during the day, the nitrogen dioxide photolyzes, recreating NO and the triplet monatomic oxygen radical O(³P) (reaction 2). The latter product reacts with oxygen to create O₃ (reaction 3). The fourth reaction is known as ozone titration, and can lead to ozone removal. While present at all times, the daytime photolysis of NO₂ favours ozone formation, and this reaction is the main route by which ozone is created during the day. Reducing photolysis rates alone will thus reduce ozone production. We have done tests without the turbulence parameterization where the photolysis rates were adjusted by the vertically weighted proportion of the lowest (original) model layer within the canopy (i.e. the mixing within the canopy layers is assumed to be instantaneous), and found that the light level alone can account for a significant portion of the ozone reduction seen with the full model. The addition of the turbulence parameterization serves to further separate the chemical regimes (above versus below canopy), enhancing the light effect. The reduction in diffusivities also results in more trapping of surface-emitted NO (at all times) – so a secondary effect, which also reduces ozone, is via reaction (4). This effect will dominate at night (since the light levels then are zero anyway), though the bulk of NO emissions at the surface occur during the day.

We have included more discussion of the stability side of the parameterization in the revised manuscript, including the reorganization of the formula to show their derivation, in response to the Reviewer’s short comment on equations 8 and 9. Shaw et al (1988) segregated his data into neutral and stable conditions, but did not make measurements under highly unstable conditions. Lacking these observations, we assumed under unstable conditions a small extrapolation of the ratio of above to below canopy σ_w from neutral conditions, leaving the main effect of stability up to the driving meteorological model. We also added the following text:

“As noted above, the meteorological model’s predicted stability influences its predicted diffusivity coefficients. These diffusivity coefficients are used to normalize the derived diffusivity profile within the parameterized canopy layers via equation (2). The impact of large-scale stability is thus incorporated into the parameterized diffusivities within the canopy. The second variation on the parameterization, described in equations (6) through (9), attempts to account for the additional influence of stability on the profile of the variance in Eulerian vertical velocity within the canopy itself⁵⁵. For this second variation, we note that few if any observations of σ_w aggregated for highly unstable conditions exist; our “unstable” profile is thus similar to the neutral profile, with the bulk of the changes associated with unstable conditions being the result of the resolved scale meteorology’s diffusivity values. Further observational evidence is needed to improve the method for highly unstable conditions.”

Personally, I think the authors are correct that the impact of within-canopy processing of reactants is an essential component currently hindering accurate prediction of regional to global reactivity. However from what has been presented in the manuscript, I’m not convinced that the proposed strategy adequately brings current understanding to bear on the problem, and therefore that their improvements in surface O₃ biases arise for the right reasons.

We are very aware of the limitations of our methodology, and the need for improvements! As with any idea that brings together different branches of science, approximations have to be made, and the details within any of those can, and should, be called into question. Our hope is that this work stimulates an ongoing discussion, as well as additional measurements (e.g. σ_w values under unstable conditions, work

on extensions of the Harman and Finnigan (2007) approach to capture the lowest half of the region under the canopy height, improvements to satellite retrieval algorithms in order to improve canopy heights and LAI values on the large scale, etc.). We acknowledge some of our methodology's limitations within the revised summary paragraph of the main body of the text: "Our results are based on a single month's model simulation; longer simulations are planned. Further improvements to the parameterization may be possible with better forest classification, leaf-area and clumping index data. Our approach for parameterizing within-canopy turbulence has been used in past one-dimensional models (Makar et al,1999; Stroud et al,2005; Gordon et al, 2014), but we note that the approach can't simulate some aspects of that turbulence (e.g. counter-gradient fluxes (c.f. Finnigan, 2000)). Improvements to the manner in which within-canopy turbulence is parameterized within regional chemical transport models should thus be a focus for future work, along with measurement campaigns to improve on these parameterizations, and improvements to satellite retrieval algorithms used to recover the necessary model inputs for the parameterizations. Furthermore, we do not rule out other potential contributing factors to the ongoing positive ozone bias (such as errors in input emissions data). We have found in other work (Makar et al, 2016) that sub-grid-scale variability (the extent to which small towns and villages within forested regions are resolved by a given model resolution) – can also significantly affect the canopy parameterization results; applying the parameterization to higher resolutions may also be beneficial. Our methodology should be treated as a first stage in what we hope will be an increased research focus on forest processes and their influence on regional and global chemistry.

Despite these limitations, our results show that the forest environment has a significant influence on atmospheric chemistry near the surface of the earth, and are in accord with observations at forested sites. The resulting decreases in ozone have been shown to encompass a very large extent of North America, and will likely impact regions with significant forest canopies (from Figure 3(c), these include the northern boreal forests of North America and Russia, the world's rainforests, and smaller regions such as south-eastern Europe, south-west Asia, and Japan). "

Specific comments:

Line 54: '... the effect is most pronounced...' What effect is most pronounced?

The sentence, "Seasonally, the effect is most pronounced in the summer¹⁴," has been changed to "Seasonally, these positive biases are the most pronounced in the summer¹⁴."

Lines 69-71: 'The cause of this reduction is ...' How do the authors know that light attenuation is the cause? How have they eliminated a change in the emissions? or canopy-induced turbulence modifying the likelihood that reactants find each other to react (i.e. segregation via organized turbulent motions with timescales similar to reaction timescales)?

The sentence in the original manuscript reads, "The cause of this reduction in light intensity is the increased attenuation of light due to the presence of the leaves from deciduous vegetation, present only from spring through fall." The point is that "more leaves means more light reduction". The reviewer has apparently mis-interpreted the intent; no change to this sentence.

Line 78: ‘... are consistent with ...’ I agree that a reduction in light levels is consistent with the observed decreases in O₃, but the authors have not demonstrated that modification of the light regime is necessarily responsible. As above, how do the authors eliminate the many other potential processes that could also be responsible?

We were not trying to eliminate other possibilities here, rather, make the case for one of those possibilities being canopy shading and forest turbulence. The sentence has been changed to: “Reductions in light levels associated with foliage^{21,22} are thus one possible cause for the observed decreases in ozone of Figure 2(a), consistent with known ozone chemistry.”

Lines 88-91: Most of the research I’ve encountered suggests that within and above tall vegetation vertical scalar fluxes are larger for a given vertical gradient compared to over smaller roughness or over bare soil (e.g., Raupach 1979), which implies larger scalar diffusivities over tall canopies. Please explain.

Yes, the scalar diffusivities over larger canopies will be larger than for smaller canopies. Our point here is that the current assumption is that the details of the profile in diffusivity, i.e. the “shelf” in diffusivities associated with forest canopies, (where, starting up from the ground, the sigma values (and K’s) at first increase slowly, then rapidly increase, before levelling off again) does not matter. We have changed the part of the sentence reading “combined with vertical diffusivity reductions” to “combined with foliage-induced modifications to vertical diffusivity”

Line 164: ‘... one-sided leaf area ...’ I suspect that the mixing efficiency (and therefore the within-canopy residence time and light regime) of reactants emitted beneath and within a canopy with a dense overstory (e.g., a deciduous forest where emissions occur primarily in the upper canopy) will certainly mix differently than one where the canopy is most dense near the surface (e.g., a pine forest, where most emissions are biased toward the surface but are more vertically distributed). So, how do the authors account for the influence of the vertical variation of the canopy elements?

As noted in the original methodology description, profiles of LAI by vegetation class were combined with satellite derived total LAI; these and the standard emissions algorithms were used to emit biogenic hydrocarbons into individual layers in the model. It’s worth pointing out that this is a departure from the usual flux boundary condition, though. The original sentence reads, “Attenuation was also applied to PPF_D using the BEIS3 emissions algorithms²; downward cumulative LAI values and the LAI distributions were used to determine the PPF_D penetrating to each layer, and the mass of biogenic emissions from each layer.” The revised sentences reads: “Attenuation was also applied to PPF_D using the BEIS3 emissions algorithms²; downward cumulative LAI values and the LAI distributions were used to determine the PPF_D penetrating to each layer, and the mass of biogenic emissions from each layer. These were emitted directly into the given layers rather than as a surface-based mass flux, aside from NO emissions, which are derived from soil sources, and remained as a surface flux.” Regional chemical models usually include biogenic emissions as a flux boundary condition on the diffusion equation; adding the mass to the canopy layers is more realistic.

Line 169: What do the authors mean that they’ve ‘assumed the attenuation of diffuse light to be relatively independent of wavelength’? Is the attenuation independent of wavelength or not? If it’s only ‘relatively independent’, then I presume there is some portion that is dependent on wavelength. How is this dependency treated?

The sentence has been modified to “assumed to be independent of wavelength”.

Lines 179-180: How do the authors define a ‘forest canopy’ here?

The criteria for defining a forest canopy were discussed in the section entitled “Criteria for using a canopy parameterization.” The sentence has been modified to end “grid cells contain forest canopies (described in more detail below).”

Lines 181-183: What about light-induced vertical variation of reactant emission?

Included. See response to comment about line 164.

Lines 184-186: What is a ‘process stream’?

The phrase “process stream” has been replaced with “two sets of model processes (one for all columns containing forest canopies, and one for all columns without forest canopies)”

Lines 217-219: Please see my general comment above discussing M-O over tall canopies.

Please see our response, above.

Line 238: Eq. 1 appears to be an exponential function. To me, an ‘inflection point’ implies a change in concavity (i.e. a location where the second derivative of a function = 0). Perhaps I’m missing something, but, how does an exponential function have multiple inflection points?

The phrase “main inflection points of equation (1)” has been changed to “main changes in canopy vertical diffusivity and leaf area index in the vertical dimension (the latter influencing equation (1) at each layer)”

Eq 5 and Lines 268-270: A couple questions: 1) a better fit to which ‘observations’? and 2) The authors comment on a ‘flattening of σ_w with increasing stability’. What about with increasing instability? I ask this because I saw a recent manuscript (Patton et al. 2016) showing σ_w to increase with height above the canopy with increasing instability.

- (1) The observations of Patton et al (2003) and Shaw et al (1988), the references used in the original sentence.
- (2) The sentence could just as easily have been stated “increasing gradient in σ_w with increasing stability”; i.e. we’re saying the same thing. Note that we’ve re-written this section in response the next comment.

Eqs 8 and 9: Where to all these parameters come from? Are they just made up? Are the heights at which they switch independent of a particular canopy density distribution? How is this independence justified?

As noted in the original manuscript, they were a fit to the observations of Patton et al (2003) with the addition of a stability dependent flattening of the profile to follow the observations of Shaw et al (1988). We assume that the reviewer’s comment means that more detail in the description is desired. We’ve

modified the section, given that Nature Communications has a larger page limit than Nature, to allow us to better describe the procedure, and we've included our own caveats on the use of the formulae, viz:

"... with increasing stability (Shaw *et al*, 1988). Stability is controlled at scales larger than the canopy by the driving meteorological model, which in turn sets the magnitude of the diffusivities via the ratio expressed in equation (2). The key concern with equations (5), and (6) through (9) which follow, is that they adequately represent the shape of the resulting profile in σ_w , hence the profile of diffusivity, which in turn is normalized by the meteorological model's diffusivity at the resolved scale.

Our approach was to follow the available observations (Shaw *et al*, 1988) of σ_w profiles to the greatest extent possible. First, we noted that the σ_w/u^* profile of equation (5) (Raupach, 1989) when overplotted with the data from multiple observation studies in presented in Patton *et al* (2003) and Raupach *et al* (1996), tended to overpredict the magnitude of σ_w below the canopy height. The coefficients of (5) were adjusted to create a σ_w/u^* profile centered in the distributions of the available these observation data (Patton *et al* (2003), Raupach *et al* (1996)). The resulting equation for this observation-centered σ_w/u^* profile was

$$(-0.1 \leq h_c/L < +0.1) :$$

$$\sigma_w \left(\frac{z}{h_c} \right) = \begin{cases} 1.0 u^* & , \quad \frac{z}{h_c} > 1.25 \\ u^* \left[0.625 + 0.375 * \cos \left(\frac{\pi}{1.06818} \left(1.25 - \frac{z}{h_c} \right) \right) \right] & , \quad 0.175 \leq \frac{z}{h_c} \leq 1.25 \\ 0.25 u^* & , \quad \frac{z}{h_c} \leq 0.175 \end{cases} \quad (6)$$

For the GEM-MACHv2.1 simulations, equation (6) was used for neutral atmospheres ($-0.1 \leq h_c/L < +0.1$), given that the corresponding suite of observations were for neutral atmospheric conditions. Observations of $\sigma_w/U(34m)$ were segregated into neutral and stable environments were available from one measurement study(Shaw *et al*, 1988). These profiles showed that above the canopy, the rate of increase in $\sigma_w/U(34m)$ with decreasing stability (i.e. going from stable to neutral conditions) was much larger above the canopy than near the surface. Ratios of the values of $\sigma_w \left(\frac{z}{h_c} = 1.8 \right) / \sigma_w \left(\frac{z}{h_c} = 0.3 \right)$ for stable and neutral conditions were constructed from these data, resulting in ratio values of 4 and 2.5 for stable and neutral conditions, respectively. The neutral atmosphere value of this ratio from (6) is 4.0, conforming to these observations. For stable atmospheres, we noted that the trend of the flattening of the σ_w with increasing stability shown in Shaw *et al* (1988) would imply a flat profile (no variation in σ_w with height) for stability levels with $h_c/L > 0.9$. We therefore set the slope of σ_w to u^* to be constant for very stable conditions:

$$h_c/L \geq 0.9 : \quad \sigma_w = 0.25u^* \quad (7)$$

In order to have a smoothly varying change in the profile shape between neutral and very stable atmospheres, equation (6) was modified so that the coefficients determining its vertical variation would asymptotically approach the neutral atmosphere solution at $hc/L = 0.1$, and the very stable atmosphere solution at $hc/L = 0.9$:

$0.1 \leq h_c/L < 0.9 :$

$$\sigma_w\left(\frac{z}{h_c}\right) = \begin{cases} 0.25 \left(4.375 - 3.75 \frac{h_c}{L}\right) u^* & , \quad \frac{z}{h_c} > 1.25 \\ u^* \left[A + B * \cos\left(\frac{\pi}{1.06818} \left(1.25 - \frac{z}{h_c}\right)\right) \right] & , \quad 0.175 \leq \frac{z}{h_c} \leq 1.25 \\ 0.25 u^* & , \quad \frac{z}{h_c} \leq 0.175 \end{cases} \quad (8)$$

where

$$A = 0.125R + 0.125$$

$$B = 0.125R - 0.125$$

$$R = 4.375 - 3.75 \frac{h_c}{L}$$

We note that this formula provides an above – canopy to surface level σ_w ratio of 0.5 when $h_c/L = 0.5$, in accord with the available observations (Shaw et al, 1988).

The available data (Shaw et al, 1988) categorized the change in σ_w for stable (defined in that reference as $h_c/L > 0.5$) and neutral ($-0.1 \leq h_c/L < +0.1$) environments, but no data were available for unstable environments. The observed change in the above-canopy to surface ratio in σ_w from stable to neutral conditions implies that the ratio may increase further with further decreases in stability. Rather than extrapolate far beyond the available observations, we chose a ratio of above – canopy to surface σ_w of 5.0 for unstable conditions, constructing the final of the four profiles using this assumption:

$h_c/L < -0.1 :$

$$\sigma_w\left(\frac{z}{h_c}\right) = \begin{cases} 1.25 u^* & , \quad \frac{z}{h_c} > 1.25 \\ u^* \left[0.75 + 0.5 * \cos\left(\frac{\pi}{1.06818} \left(1.25 - \frac{z}{h_c}\right)\right) \right] & , \quad 0.175 \leq \frac{z}{h_c} \leq 1.25 \\ 0.25 u^* & , \quad \frac{z}{h_c} \leq 0.175 \end{cases} \quad (9)$$

This last assumption may result in an underestimate of the “trapping” of pollutants in unstable environments if the relative ratio in above-canopy to surface σ_w values is greater than 5.0. Additional observations under unstable conditions are needed to improve this version of our parameterization further. Also, the intent of the profiles is to create the shape of the profile of diffusivity with the assumption that the diffusivity at the resolution of the driving meteorological model is correct. From the spread of observed σ_w values (Patton et al, 2003; Raupach et al, 1996), further refinements based on more specific types of vegetation are possible.”

Lines 291-295: How have the authors deemed these diffusivity profiles to be ‘reasonable’? Has there been some attempt to connect these parameterized profiles with the Borden observations?

The sentences have been changed to “, attempt to account for canopy-specific changes in atmospheric stability (Shaw et al, 1988). The subsequent scaling of the resulting profile to meet the diffusivity of the driving meteorological model at height z_1 thus allows an approximation to the diffusivity within the canopy based on the available observations of σ_w , while taking into account differences associated with changes in canopy height, friction velocity and the response of canopy turbulence to atmospheric stability.”

Given the variation in σ_w values in Patton et al (2003) and Raupach et al (1996) across different forest types a one-location comparison might not be that useful; the performance of the model over the larger domain is a more relevant test. We did, however compare the above and below canopy ozone concentrations from the model to those observed at Borden for the same time period (see discussion above).

Reviewer#3:

Air quality modelling, especially at regional scale, suffers an historic lack of visibility although it is intrinsically multidisciplinary encompassing physical, chemical and mathematical modelling along with emission estimates, land use retrieval and meteorology. The paper reviewed here might help reaching out the broader geophysical modelling community.

The paper is overall neat and well written, and clearly fits the scopes of the journal. I found a few aspects that, in my opinion, require further attention, and that I invite the authors to consider in order to make the results more robust and their exposure clearer.

Once the necessary changes have been made and questions addressed, I see no objection for accepting the manuscript for publication in Nature Communications.

We thank the reviewer for his supportive comments – our detailed responses to the general remarks follow.

General remarks

1. The large (sometime dominant) influence of emission uncertainty, anthropogenic and natural NO_x, as well as isoprene is not properly highlighted, in my view. The authors should make an effort to explain why the high bias they observe cannot be explained by the error in the emissions. For example, there is high bias in the 'Central USA' region (figure 1a), although figures 2d and 2e show that the region is not forested.

Actually, the reviewer's assertion that Figures 2(d) and 2(e) of the original manuscript show that the region is non-forested is not correct. However, we can see how that conclusion might be drawn, due to our use of a different map projection in Figure 1a (Mercator projection) compared to Figures 2(d) and 2(e) (Orthographic projection). Unless one tracks the outlines of individual states between Figure 1a and the other two figures, the projection change can trick the eye the central USA region has no forests when the two figures were compared. To address this issue, in the revised manuscript we've replotted 2(d) and 2(e) (now Figure 3) with the same projection and same sub-region boundaries as 1(a). In addition, we've created the plot below for the reviewer's benefit, with the original Figure 1(a) and the same information as the original Figure 2(d) and 2(e) in the same image. The eastern half of the "central USA" region does have substantial forest canopies, over 20m in height and with leaf area index values > 4.0. With regards to the high bias, we've added a sentence in the final part of the main body of the paper noting that while the work carried out here results in improvements to ozone performance, we can't rule out other sources of model error as contributing the positive biases in existing models; viz: "Furthermore, we do not rule out other potential contributing factors to the ongoing positive ozone bias (such as errors in input emissions data)."

Figure R3.1 Comparison of (a) original model biases, (b) canopy heights, and (c) leaf area index values, plotted with the same map projection.

On the same line, literature results show that RMSE error for modelled ozone is similar in Europe and North America, although the forest impact in Europe might be much smaller, according to Figure 2f. In my view, while in Europe that of the emission's error is a big issue, it might not be the case in North America, but it cannot be excluded from the discussion as it stands. I invite the authors to comment on this point.

Emissions errors are indeed one of the main concerns for regional modelling, given that the accuracy of emission reporting is always in question. Regulatory agencies make every effort to obtain accurate emissions information, and the emissions inventories are improving with time. As noted above, we have acknowledged this in the revised manuscript.

However, we also feel that it is important not to focus solely on the emissions part of the problem, since other factors, such as the process we've examined, may have an equal or bigger impact. For example, consider the following comparison of June-July-August average hourly ozone concentrations from observations and one model from an ensemble study (the AQMEII-1 model intercomparison for Europe, Solazzo et al, 2012, Figure 9 (a,d)), juxtaposed with the canopy heights from satellite retrievals for Europe (Figure R3.2, below).

Figure R3.2: Comparison of European JJA O3 MSE from Solazzo et al, 2012 with the position of forest canopies in Europe.

This particular model in the ensemble clearly has a high ozone bias (compare middle panel to panel to the observations panel on the left in Figure R3.2) – however, comparison to the forest canopy heights in the right-most panel of Figure R3.2 suggests that the spatial location of some of the largest biases (Northern Italy, north-eastern Spain, Switzerland, Austria, the borders between Poland and Germany, the Czech Republic, and Slovakia, southern Germany) correlate well with the spatial locations of the deepest forests (hence most likely to have lower ozone concentrations than predicted, were canopy processes to be included in the model). Consequently, biases might be reduced through the incorporation of forest effects.

A similar plot this time comparing the mean square error for long-term timescales (Solazzo and Galmarini, 2016, Figure 3(b)) to European canopy heights appears below (Figure R3.3). The given reference demonstrated that model error may be analyzed according to time scale, and that much of the mean square error resides in the longest (“Long Term : LT”) timescale. In this particular example, the largest errors associated with the COSMO-Muscat model are once again located in regions with significant forest canopies.

Figure R3.3. A comparison of Solazzo and Galmarini, 2016 Figure 3(b) May through September mean square errors for ozone with canopy heights for Europe. ID = intra-day, LT=Long Term, DU=diurnal, SY=Synoptic time scales.

While the above comparisons do not rule out the possibility of emissions inputs or other model errors contributing to the model bias and mean square errors, they do suggest that improvements to European predictions of ozone could be *also* be achieved through the use of forest canopy parameterizations such as the one we have constructed, and, consequently, that forest processes may have a significant impact on European ozone.

We have added the following sentence to the manuscript:

“Multi-model ensembles of European ozone predictions (Solazzo et al, 2012; Solazzo et al, 2016) compared to the same global forest canopy satellite data have a similar correspondence between positive ozone biases and forest locations (not shown).”

2. The influence of forest canopy on ground-based monitoring stations is not clear to me, since the stations are not sited within the forest. Within the canopy the vertical mixing is inhibited and concentration much lower than aloft, but how that information is carried to the monitoring station, outside the forested area, is not clear. Somehow, the authors should explain that monitoring stations falling within model grid cells interested by the canopy parameterization have a spatial representativeness that includes the effect of the forest (if that is the case...), and that volume-averaged values (that of the models) are compared against point measurements (that of the station). Unless I have misunderstood, this point I believe is very important. Otherwise the model bias could, again, be explained

by biased-high emissions in the forested areas. Line 91 indeed says '...limits the vertical mixing of chemicals emitted near the earth's surface.' But (anthropogenic) pollutants are not emitted within the forest, but rather in the grid cell containing a fraction of forest, right? Hope I have explained my doubt.

These are very good points. For a related conference proceedings publication, we've carried out a sensitivity study changing the criteria for the canopy parameterization, which has some bearing on these issues. It's also important to point out that horizontal advective transport from forested areas to non-forested areas at the resolved scale will also lead to lower initial ozone at the emissions regions, hence lower total ozone levels. As a result of horizontal transport, the forested and non-forested regions should not be considered as separate entities – conditions in the first will affect conditions in the second, regardless of the grid resolution issues raised by the reviewer. We've included the following into the text of the revised paper, in the initial discussion, to address these issues:

“We note that a key issue for air quality models is that of spatial representativeness – the ability of a model to distinguish small features in the domain is limited by the available computation resources, and current large-domain air-quality models have resolutions typically on the order of 2.5 to 30 km. As a result of these resolution limitations, observational datasets must be compared to average conditions over regions between 2.5x2.5 and 30x30 km². The extent to which the observations and model values match will thus depend in part on the extent to which the upwind footprint of the observations represents the actual conditions within the region encompassed by the model grid-cell, as well as on the extent to which the model input databases represent the vegetation characteristics within that grid cell. All chemistry and transport processes, including the forest mixing and shading examined here, thus represent average conditions within the smallest volume resolvable by a model. Despite issues of spatial representativeness, transport of low concentration ozone from forested to non-forested regions will also impact total ozone levels in the latter. With horizontal advection, low ozone concentrations in forested regions surrounding a city may thus reduce ozone levels in the latter.”

We have also added the following sentence towards the end of the article:

“We have found in other work (Makar et al, 2016) that sub-grid-scale variability (the extent to which small towns and villages within forested regions are resolved by a given model resolution) – can also significantly affect the canopy parameterization results; applying the parameterization to higher resolutions may also be beneficial”.

3. Merging the previous two comments, I invite the authors to show in figure 3 only the results at stations falling within model grid cells interested by the canopy parameterization. The other stations should not report any change, right? Or maybe add contour lines for the 'zero'.

We have added an additional figure (new 4(b)) and some discussion to show the impact of the canopy parameterization at all of the model gridpoints throughout the model domain (canopy parameterization hourly ozone minus the base case hourly ozone), to show the full spatial extent of the ozone changes, at all model gridpoints (as opposed to just at the observation stations) . This figure shows the large spatial

extent of the changes associated with the canopy parameterization, both within and outside of urban areas. The new text added follows:

“The change in the average ozone concentration at every model grid-cell is shown in Figure 4(b). The canopy parameterization has an impact throughout the model domain, usually resulting in ozone decreases, with the greatest decreases in the California and south-eastern USA forests, and lower decreases in the boreal forests of Canada. Ozone increases in Los Angeles, and slight increases may be seen in other areas. The impact of the canopy parameterization is widespread, and covers regions outside of the regions with high leaf area index and canopy heights of Figure 3(a,b), such as the central plains of North America, and the downwind area of much of the Atlantic. This demonstrates that downwind transport of low-ozone air from canopy to non-canopy regions is sufficient to decrease ozone levels in the latter.”

Specific comments (in no particular order)

- I believe it should be said clearer that conclusions are entirely drawn based on one month model simulation.

Done. A sentence to this effect has been added early in the discussion of the revised manuscript: “Our results are based on a single month’s model simulation; longer simulations are planned.”

- Any effort made to simplify the exposure of the method and results would be beneficial. For example I would invite the authors to consider changing the naming of the modelled cases to 'base' and 'base+canopy', and to produce results for one model version only. I can't see the advantage of having version 2 and version 2.1, since the focus of the study is the canopy parameterization. I believe this doubling of the results can bring confusions and distract from the main goal, especially considering that the two versions produce similar statistical scores.

The additional parameterization asked the question “To what extent does the available data on the response of canopy turbulence to larger scale atmospheric stability affect the parameterization results?”, an issue raised by our second reviewer. In Table SI3 in the original manuscript, the stability-dependent parameterization resulted in a larger decrease in the mean bias of ozone relative to its own base case than before. Yes, the two “canopy” runs had similar results – but they should be compared to their respective base cases. The base case model for v2 had a mean bias of 4.26 ppbv, while that for v2.1 had a positive mean bias of 6.29ppbv. Consequently, the first canopy parameterization reduced the North American mean bias by $(4.26-1.76=)$ 2.5ppbv, while the second canopy parameterization reduced the North American mean bias by $(6.29-1.78=)$ 4.51 ppbv. The information gained is thus that the impact of within-canopy turbulence may be relatively large, given that the change in parameterization resulted in a greater absolute total change in mean ozone levels.

We’ve added the following text to clarify the intent of the second test and its outcome to the reader: where we describe our evaluation of the hypothesis, we have added the sentence: “The second of these

canopy simulations investigated the extent to which changes to the shape of the expected profile of vertical diffusivity in response to stability might impact model performance.” We’ve also added the sentence in the discussion on the statistics results: “The two different canopy vertical diffusivity parameterizations also resulted in different levels of absolute decrease in the North American ozone mean bias; 2.50 ppbv for the v2 parameterization, and 4.51 ppbv for the v2.1 parameterization, suggesting that the shape of the within-canopy profile of the variance in vertical velocity in response to larger scale atmospheric stability may have a significant impact on surface atmospheric chemistry.”

- Line 111. '*...has significantly reduced the ozone bias in the forested regions...*'

We’ve used “has significantly reduced the ozone bias in many of the regions with the highest positive bias in the original simulation, due to the prevalence of dense vegetation in those regions”. Note also the modification of the figures so that the reduction in bias at station locations plot is now matched by a change in model average ozone concentration plot, to show the spatial extent of the changes in ozone.

- Line 138-139. *The correlation coefficient not changing is somehow expected, as the new canopy module only acts on the bias, and the correlation is bias-independent. However, it shows that the new parameterization does not feed the model new covariance error, in light of the bias-variance trade-off theorem (Krogh, A., Vedelsby, 1995). Just a comment.*

- *Although not fully justified, the use of parametric statistics is customary in air quality model evaluation. I would encourage the authors to use the following three metrics: squared bias (or mean bias) for distance, squared ($var(obs) - var(mod)$) for variance, and $(1-r)\sigma_{mod}\sigma_{obs}$ for covariance (σ is the standard deviation and var the squared standard deviation). The summation of the three returns the total MSE. All other (parametric) metrics are derived from these three (see Murphy et al 1988; Gupta et al 2009) and are redundant.*

The formula for variance and covariance above are different from what we’ve seen in the literature; e.g. Solazzo and Galmarini 2016, equation (6) has the variance defined as

$$VAR = (\sigma_M - \sigma_O)^2$$

And the covariance defined as:

$$COV = 2(1 - R)\sigma_M\sigma_O$$

The above two definitions have been added to the list of statistics in Methods along with the given reference, and have been included into Table 1 which describes model performance. The canopy models have better performance for variance than their respective base case models, while slightly worse performance for covariance.

The values of σ_M and σ_O have been included in the table

- Eq (1). I am no expert, but I believe that this equation would later need an ad-hoc evaluation.

While based on a fundamental physical relationship (Beer's law), equation (1) is based on direct observational evaluation in the references quoted in the original manuscript (Nilson, 1971; Monsi and Saeki, 1953), and appears in multiple references throughout the forest measurement literature. Further evaluation at this point should not be necessary.

- Item 2, Line 171-177. I agree with the statement here. Please consider reading Finnigan (2000); Cionco (1965).

We're unsure as to why the reviewer agrees with the statement but added the additional references. Perhaps a typo and "disagree" was intended? Finnigan (2000) used the same reference as Patton et al (2003); a suite of observations compiled by Raupach et al, (1996) – we've included that underlying reference in the revised version of the manuscript. We've also modified the statement as follows:

"(2) Turbulence parameters such as the variance in the Eulerian vertical velocity, when plotted with a vertical coordinate scaled to the canopy height and scaled by parameters such as the friction velocity, show a remarkable similarity in profile shape across different vegetation types and canopy heights (Patton et al, 2003; Finnigan (2000); Raupach et al (1996)). The localized near-field theory first proposed by Raupach (1989) and adopted to our 1D canopy models (Makar et al (1999), Stroud et al (2005) and Gordon et al (2014)) has been used here to approximate turbulent mixing throughout the canopy. We link the resulting diffusivity profiles to the resolved scale by their normalization to the above canopy diffusivity of the driving meteorological model. The approach and its limitations are discussed below (Coefficients of Vertical Diffusivity)."

- Line 232. Were the urban areas detected by using the modeled heat flux? How reliable is that? Would not be feasible to use imagery analysis?

The population and heat flux were actually linearly related in the model inputs, in that the urban heat flux itself was created using per capita heat fluxes for North America from a population field used for spatial allocation of specific emissions fields, following Makar et al (2006). Rather than confuse the issue, we've modified the manuscript with the equivalent population per 10km grid-square as the relevant criteria (50,000).

- Line 258. What about the lateral exchange at the edge of the forest (in light of main comment #2)?

See our response to the comment above. We have modified the original figure to also show the change in model mean ozone at every model grid cell. This shows a substantial influence of the canopy through advection downwind.

- *Statistical scores. Please specify the time aggregation implied in the spatial summation over the stations.*

The title for the table gave the time aggregation for the stations (none; hourly values) . We have added the word “hourly” in the first reference to Table S13 (now moved to the main manuscript in response to the Editor’s request) in the revised manuscript.

Some editorial suggestions

Line 57. Please consider ',...measurements of ozone. concentration in the forest....differences at different heights...'

The revised sentence reads, “Parallel to these modelling efforts, measurements of ozone concentration in the forest research community have shown pronounced differences with height in dense forest canopies, suggesting a very different chemical environment exists above versus below the foliage”.

Line 59. Please provide reference to you statement ending with 'foliage'.

Three references were used in the sentences which followed. They have been moved up to the end of the given sentence.

Line 60. Please provide reference to you statement ending with 'environments'.

Same references as for line 59; Amazon rain forest, central Massachusetts, and Borden Research Station in Canada. Adding the references again here is unnecessary.

Line 61. 'O3'. For consistency should be 'ozone'.

Modified as suggested.

Line 73. What do the authors mean by 'light levels'? Is it the intensity, the squared amplitude, please specify.

The start of the sentence has been replaced with “If the light intensity becomes sufficiently low”

Line 88. '...relatively to non-vegetated...'?

Our original phrase, “relative to non-vegetated surfaces” is actually the more grammatically correct form here, so we’ve retained it.

Line 89. Please consider '...surfaces, the attenuation of downward radiation due to foliage, combined with inhibited vertical diffusivity, suppresses...'

We've changed the sentence in response to other comments to "We hypothesize that, relative to non-vegetated surfaces, foliage-induced attenuation of downward radiation, combined with foliage-induced modifications to vertical diffusivity, suppress daytime photochemistry and limit the vertical mixing of chemicals emitted near the earth's surface".

Line 99. The references for GEM-MACH have been already provided at line 41, although different ones. Please verify.

Line 99 references removed, line 41 references were corrected to 6,7,8,13 ("12" in the original manuscript was a typo, thanks for catching this).

Line 100-104. This I find confusing. According to me it should read: '...two base-case simulations to which the vegetated canopy parameterization was added, for a total of four sets of model simulations...', or something on that line.

The paragraph has been rewritten to read "We evaluated our hypothesis using our chemical transport model (GEM-MACH, versions 2 and 2.1), after modifying the model to include a canopy parameterization with two approaches for vertical diffusivity applied separately to the two model versions (see Methods). Four sets of simulations were carried out on GEM-MACH's operational 10 km resolution North American forecast domain; two "base case" simulations using different versions of the unmodified model (versions 2 and 2.1, the latter including updates for urban heat island and the Monin-Obukhov Length; see Methods), and two simulations wherein vegetated canopy parameterizations were added to the base case simulations. The second of these canopy simulations investigated the extent to which changes to the shape of the expected profile of vertical diffusivity in response to stability might impact model performance. The simulations mimicked a three day forecast procedure, comprising consecutive, linked, three day forecasts at 0 and 12UTC spanning the month of July, 2010. Statistical evaluation of the model performance in each simulation was carried out using surface network hourly observations of ozone."

Figure 3a. 'Shading' has not been defined/introduced before. Maybe better keep 'canopy'.

The caption within former Figure 3(a) (now Figure 4(a)) has been modified as suggested.
Eq(1) - LAI should read LAI(z), according to the detail given later, am I right?

To be exact, it's the total integrated LAI from the canopy height down to the level z within the canopy, so not the same as the value of LAI at a given layer. However, we've stated just before (1) that this was to the surface of the earth, so the given definition (which is the total LAI) still holds. However, we can see how this might cause confusion later in the text, so we've modified the equation and its description to read:

- (1) "The theoretical basis for the attenuation of light in vegetated canopies may be derived from Beer's Law as the probability of beam penetration (which may also be interpreted as the fractional light penetration) to a given level z below the top of the canopy, using equation 1 below^{21,22}:"

$$P(\theta, z) = e^{-\frac{G(\theta) \Omega(\theta) LAI(z)}{\cos(\theta)}} \quad (1)$$

Where $P(\theta)$ is the probability of beam penetration at solar zenith angle θ , $G(\theta)$ is the projection of unit leaf area in the θ direction (for a spherical leaf distribution, the usual assumption, $G = 0.5$), $\Omega(\theta)$ is the clumping index (a measure of the randomness of the leaf spatial distribution, $\Omega=1$ if the leaves are randomly distributed), and $LAI(z)$ is the total leaf area index (the one-sided leaf area in the column per unit ground surface area) downwards from the canopy height to the given level within the canopy. ...”

Line 179. '...to determined which model grid cells contains forest canopy'. I would add ' and what fraction'.

Good point. We’ve modified the sentence to read

“ (1) Use satellite-retrieval-derived LAI, $\Omega(\theta)$, canopy height data, vegetation fraction, and population density data, to determine which model grid cells contain vegetated canopies (described in more detail below);”

Line 209. See previous 'Line 99' comment.

The reference numbers in this case were correct, but I assume that the reviewer means that they only need be referenced once? The sentence has been revised to “The details of this on-line chemical transport model’s formulation may be found in the references provided above, along with a comparison of its performance relative to observations and peer-group models.”

References

Finnigan, 2000. Turbulence in plant canopies. Annual Review of Fluid Mechanics, Vol. 32: 519-571
Cionco R (1965) A mathematical model for air flow in a vegetative canopy. J Appl Meteorol 4: 517-522
Gupta, V.H., and et al., 2009. Decomposition of the mean square error and NSE performance criteria: implications for improving hidrological modelling. Journal of Hidrology 377, 80-91
Murphy, A., 1988. Skill scores based on the mean square error and their relationships to the correlation coefficient. Monthly Weather Review 116, 2417-2424
Krogh, A., Vedelsby, J.: Neural network ensembles, cross validation, and active learning, in: Advances in Neural Information Processing Systems, 7, 231-238, 1995

Reviewers' Comments:

Reviewer #3 (Remarks to the Author)

I believe that the questions and issues raised by the reviewers have been addressed more than satisfactorily by the authors, who have proved a deep knowledge of the scientific literature.

As I pointed out during the first review stage, I believe that the broader air quality community will benefit substantially by the new direction proposed by this work. Over the last years much effort has been spent over the chemistry part of the models (with alternate fortune, in my view). But with the proposed inclusion of a canopy module, the basis of a new, promising, research branch will be set (I hope) and it has the potential to become a breakthrough (something like the inclusion of urban effects on weather models some 15-20 years ago). In this respect, as the authors point out in their comments, the proposed parameterization needs not to be fully functional and robust at this early stage. Important is that the idea is taken on board by the air quality community and expanded.

I think, therefore, the overall work needs no further justifications and/or technical amendments as it is scientifically sound as it stands and should be accepted for publication.

I shall say, however, that with respect to the first version, this reviewed version has lost the initial agility and directness. I am not familiar with the editorial requirements of 'Nature Communications', but for any other specialized journal I would recommend to move substantial portion of the text to some sort of appendix or supplementary material. The focal point of the paper should be the new canopy module, with short, concise discussion about it. All elements distracting from this main point should be (re)moved, in my view.

For example, the discussion about the spatial representativeness could be condensed into one line – just mentioning the horizontal transport. On the same line, the figure 5 and discussion thereabout are insightful but unnecessary in the main body of the text. Possibly tables and methods could also be shortened?

One minor suggestion is to consider changing 'advection, diffusion' in line 12 with 'transport and dispersion', which, I think, better describe these phenomena at large scales.

Reviewer #4 (Remarks to the Author)

Overall assessment: The manuscript describes an effort to improve surface ozone predictions by including a canopy model in a 3D numerical simulations of atmospheric chemistry. The authors argue that including the canopy model significantly reduces the BIAS in surface ozone levels. With the current level of details, it is difficult to assess if the results are as meaningful as the authors claim. It is not clear if the canopy model improves the predictions of the large scale model or if simply refines predictions below the first grid point. This is very important, as the former is a major advance while the latter is perhaps an incremental improvement. The manuscript also fails in clearly outlining the processes that are considered in the canopy model and, more importantly, explaining how the canopy is impacting ozone levels. My main comments are summarized below.

1. A critical point that requires clarification is the comparison between ozone observations and model predictions showed in Figures 1 and 4. The changes in model BIAS between these two figures is the key result of the manuscript. Nevertheless, the reader is not given enough information to interpret the results. I am told that these are "surface ozone levels". What does that mean in a forested area? Are these values of ozone averaged in the entire vertical extent of the atmospheric boundary layer (ABL), or are these values of ozone tens of meters above the forest,

or values of ozone inside the forest (say 2m above the ground)? If the inclusion of the canopy model does reduce ozone levels and significantly reduces model BIAS in the ABL, then these are very interesting results. If the reduction in ozone levels is mostly inside and just above the canopy, then these are fairly obvious results and I believe the results are perhaps of interest to a more specialized community. In my opinion the big question is whether the canopy model improves model predictions in the model layers that exist in the simulation without the canopy, or whether it simply refines the model predictions below the first original grid point. I will try to make my point clearer. In the model without the canopy, the first grid point represents the average ozone in the lowest, say, 50m. The ozone sink (deposition, etc.) is all lumped at the ground. By adding new vertical levels as a canopy model, the sink is now represented over the entire depth of the forest, obviously resulting in lower ozone levels within the now resolved forest. Is this the main/only advantage of the canopy model?

2. Figure 5e,f – “While the scatter about the line of best fit is the largest in Figure 5(e,f), the canopy model shows a significant capability to simulate the decrease in ozone compared to the original model (where the lowest layer is assumed homogeneous, that is, the difference is always zero).” The authors are perhaps too optimistic here? This is not about the scatter, but the fact that the model under predicts the ozone attenuation by a factor of 3, on average.

3. Figure 6 – When the authors compare reduction due to canopy models with other reductions of policy-relevant scenarios for climate change, it is once again crucial to clearly discuss the spatial (vertical) extent of these reductions. At the moment, it is not clear to me how to assess these results. It is possible that the reduction associated with climate change occurs in similar proportions within the entire extent of the ABL and that the canopy reductions only occur very close to the ground. This would certainly impact how these results are discussed. More details are needed.

4. I find that the manuscript falls short on explaining how the canopy model is actually reducing ozone levels. First, two effects are mentioned: reduction of photolysis rates and modifications in vertical transport. Which one is the most important? How much does each effect contribute to the total changes? Second, how does reduced turbulence diffusivity impacts ozone levels? What is the model actually doing inside the forest? Is there a vertically resolved model for ozone dry deposition and stomata uptake? If the model reduces ozone and I am not told what the physical/chemical process causing the reduction actually are, I am not sure I have learned anything new.

5. The “parameterization” of the turbulence inside the canopy is anything but elegant. It makes use of a large number of fitted coefficients, with many decimal places, while observational data in canopies shows very large variations from case to case (hardly justifying the choices made by the authors). In particular, the authors point out the importance of LAI, and then use a parameterization that does not even account for the effect of LAI on the turbulence (the effects of LAI on the in-canopy turbulence is one of the very clear effects that the community has been able to establish). Under these conditions, why not adopt a simple model based on physics (such as the Massman and Weil 1999 model, that includes effects of LAI and leaf area density in a simple and elegant way that respects many dynamical constraints) instead of an empirically fit model that has very little physics embedded into it?

6. Line 81 – “suggest that vertical diffusivity in the under-canopy region will be greatly reduced, partially decoupling this region from the rest of the atmosphere, and increasing the relative influence of chemistry on ozone formation and destruction below the foliage.” Please elaborate on how reduced diffusivity would impact these processes. My first reaction is that less ozone would be transported into the deeper layers of the forest, but it is not clear how this would impact ozone formation and destruction.

7. Line 191 – “The forest canopy parameterization also improves the root mean square error,

mean gross error, coefficient of efficiency, index of agreement and variance." This is again a bit of an overstatement. I am actually puzzled by how little difference the canopy model makes in all statistics other than the BIAS.

Response to the Reviewers

Reviewers' comments:

Reviewer #3 (Remarks to the Author):

I believe that the questions and issues raised by the reviewers have been addressed more than satisfactorily by the authors, who have proved a deep knowledge of the scientific literature.

As I pointed out during the first review stage, I believe that the broader air quality community will benefit substantially by the new direction proposed by this work. Over the last years much effort has been spent over the chemistry part of the models (with alternate fortune, in my view). But with the proposed inclusion of a canopy module, the basis of a new, promising, research branch will be set (I hope) and it has the potential to become a breakthrough (something like the inclusion of urban effects on weather models some 15-20 years ago). In this respect, as the authors point out in their comments, the proposed parameterization needs not to be fully functional and robust at this early stage. Important is that the idea is taken on board by the air quality community and expanded.

We would like to thank the reviewer for this comment, particularly the parameterization itself – we are not claiming that the parameterization is the best possible – but we are sincerely hoping that this will kick-start a serious discussion in the three research communities we are trying to bring together, in order that the importance of the canopy effects is recognized, and that efforts are made to improve on the basic idea we've set out. There are many different ways of approaching both the light attenuation and turbulence part of this work – what we have attempted here is a relatively simple approach, one which has low computational constraints and is relatively easy to implement. We are certain that better approaches are possible – our hope is that this work will spark the interest and the multi-year collaborations; the comparison of different methodologies and the collection of additional observation data needed to improve the basic idea further. A typical weather forecast or air quality forecast model has multiple options for the processes in the atmosphere – some, like ours, are based on best fits to the available observations, while others are based on underlying physical theory. The constraint on which a given parameterization is used often depends on numerical methods used and the computational resources available. For example, the approach used in large eddy simulation models has the advantage of being based on the fundamental equations of the atmosphere. However, this approach requires a very high resolution computational domain and small time-steps in order to

accurately integrate the system of equations, and extending this sort of spatial and time resolution for the much larger domains needed in regional scale weather and air-quality models is not practical with the current state of computational power. These approaches, however, can provide a benchmark for parameterizations such as the one we have used here, and may be used to improve on them.

I think, therefore, the overall work needs no further justifications and/or technical amendments as it is scientifically sound as it stands and should be accepted for publication.

I shall say, however, that with respect to the first version, this reviewed version has lost the initial agility and directness. I am not familiar with the editorial requirements of 'Nature Communications', but for any other specialized journal I would recommend to move substantial portion of the text to some sort of appendix or supplementary material. The focal point of the paper should be the new canopy module, with short, concise discussion about it. All elements distracting from this main point should be (re)moved, in my view.

For example, the discussion about the spatial representativeness could be condensed into one line – just mentioning the horizontal transport. On the same line, the figure 5 and discussion thereabout are insightful but unnecessary in the main body of the text. Possibly tables and methods could also be shortened?

One minor suggestion is to consider changing 'advection, diffusion' in line 12 with 'transport and dispersion', which, I think, better describe these phenomena at large scales.

The changes described by the reviewer were in response to requests for clarifications to the text and additional information required by all reviewers – we feel that these have significantly improved the manuscript. Some tables were moved from a Supplemental Information section to the METHODS section following the Editor's suggestion that *Nature Communications* prefers such information to appear in METHODS when possible. As part of the current round of revisions, the paper was streamlined and compared to *Nature Communications* checklist – this resulted in a more succinct Abstract, as well as Introduction, RESULTS, sections. These fall within the guidelines set by *Nature Communications*, and we feel the resulting manuscript "reads well". We note that the manuscript now has 75 references, slightly above the guideline of 70 references; given that these were added in response to reviewer comments, we hope that these will be allowed by *Nature Communications*.

Reviewer #4 (Remarks to the Author):

Overall assessment: The manuscript describes an effort to improve surface ozone predictions by including a canopy model in a 3D numerical simulations of atmospheric chemistry. The authors argue that including the canopy model significantly reduces the BIAS in surface ozone levels. With the current level of details, it is difficult to assess if the results are as meaningful as the authors claim. It is not clear if the canopy model improves the predictions of the large scale model or if simply refines predictions below the first grid point. This is very important, as the former is a major advance while the latter is perhaps an incremental improvement. The

manuscript also fails in clearly outlining the processes that are considered in the canopy model and, more importantly, explaining how the canopy is impacting ozone levels. My main comments are summarized below.

1. A critical point that requires clarification is the comparison between ozone observations and model predictions showed in Figures 1 and 4. The changes in model BIAS between these two figures is the key result of the manuscript. Nevertheless, the reader is not given enough information to interpret the results. I am told that these are “surface ozone levels”. What does that mean in a forested area? Are these values of ozone averaged in the entire vertical extent of the atmospheric boundary layer (ABL), or are these values of ozone tens of meters above the forest, or values of ozone inside the forest (say 2m above the ground)? If the inclusion of the canopy model does reduce ozone levels and significantly reduces model BIAS in the ABL, then these are very interesting results. If the reduction in ozone levels is mostly inside and just above the canopy, then these are fairly obvious results and I believe the results are perhaps of interest to a more specialized community. In my opinion the big question is whether the canopy model improves model predictions in the model layers that exist in the simulation without the canopy, or whether it simply refines the model predictions below the first original grid point. I will try to make my point clearer. In the model without the canopy, the first grid point represents the average ozone in the lowest, say, 50m. The ozone sink (deposition, etc.) is all lumped at the ground. By adding new vertical levels as a canopy model, the sink is now represented over the entire depth of the forest, obviously resulting in lower ozone levels within the now resolved forest. Is this the main/only advantage of the canopy model?

We thank the reviewer for this excellent suggestion – we had focused mainly on the evaluation of the model at the surface and noting that the effects of the parameterization could be seen far downwind in our initial analysis. Following the reviewer’s recommendation, we returned to the model output files (which included the 3D model output for both canopy and base case simulations). We constructed daily averages of the last 24 hours of each 72 hour simulation starting at 00UT for both “canopy” and “original” model simulations. We feel that the results are very significant, and have therefore included them in a new figure in the revised manuscript (new Figure 5), along with the accompanying text, along with mention of the impact in the revised abstract of the manuscript:

“We have used ozone mass ratios from the third day of successive 0 UT simulations to show the influence of the canopy in the vertical dimension. The original hourly v2.1+ model 3D ozone values were averaged on a daily and monthly basis and (canopy / base case) ratios were constructed. A cross-section through these data, located in the center of the most heavily forested part of eastern North America (Figure 4(b)), is shown in Figure 5 for the monthly mean ozone ratio (5(a)), and the daily mean ozone ratio for July 4th and July 27 (5(b) and 5(c), respectively). The figure shows that the influence of the forest canopy on ozone concentrations extends to heights far above the canopy layers or the lowest resolved original model layer, with ratios less than 1 throughout most of the atmospheric boundary layer.

Zones of lower ozone mass due to the incorporation of the canopy parameterization may be seen in all panels of Figure 5, at both the lowest model layers and between heights of approximately 850 to 690 mb (elevations of roughly 1450 to 3250 m above the surface). This overall

depression of monthly average ozone mass in the upper part of the atmospheric boundary layer (Figure 5(a)) is linked to shorter-term events in which lowest level air is transported upwards towards the top of the atmospheric boundary layer – examples of daily averages with these events may be seen in Figure 5(b) and Figure 5(c). Roughly half of the daily averaged ratios constructed showed these events along this cross-section. Isolated high ratios in these cross-sections are linked to similar events occurring further upstream. Figure 5 shows that the canopy parameterization, due to this coupling between the lowest model layer and the rest of the atmospheric boundary layer, results in a monthly average ozone mass mixing ratio *reduction* near the *top* of the atmospheric boundary layer of up to 12%, and that daily average ozone mass mixing ratio reductions in the same area may reach 40%. These effects are in addition to the reductions near the surface which may be seen in Figure 5, and are discussed and evaluated elsewhere in this work.

This comparison shows that the forest canopy has a significant influence on ozone concentrations throughout the atmospheric boundary layer, due to interactions between the resolved scale meteorology and the region encompassed by the forest canopy itself.

The canopy processes examined here may help explain a known deficiency in global chemistry models employing data assimilation of satellite column ozone to attempt to improve model performance (Parrington et al 2009). These efforts led to unwanted increases in positive ozone biases in the eastern USA, with the attribution of these effects to “errors in the ozone sources or sinks in the boundary layer mixing in the model”. Our results suggest that at least part of these errors may be attributable to the forest canopy processes discussed herein, and that data assimilation efforts to improve tropospheric ozone forecasts would improve with the inclusion of forest canopy shading and turbulence, in global and regional air-quality models.

New reference: Parrington, M., Jones, D.B.A., Bowman, K.W., Thompson, A.M., Tarasick, D.W., Merrill, J., Oltmans, S.J., Leblanc, T., Witte, J.C., and Millet, D.B., Impact of the assimilation of ozone from the Tropospheric Emission Spectrometer on surface ozone across North America, *Geophys. Res. Lett.*, 36, L04802, doi:10.1029/2008GL036935, 2009.

2. *Figure 5e,f* – “While the scatter about the line of best fit is the largest in *Figure 5(e,f)*, the canopy model shows a significant capability to simulate the decrease in ozone compared to the original model (where the lowest layer is assumed homogeneous, that is, the difference is always zero).” The authors are perhaps too optimistic here? This is not about the scatter, but the fact that the model under predicts the ozone attenuation by a factor of 3, on average.

Good point, the main impact is the bias reduction, which is shown in the first four panels (a-d) of that figure. We’ve “toned down” the text to the following (note that the former Figure 5 becomes Figure 6 in the revised manuscript: “While the main impact of the canopy parameterization is to reduce the ozone bias (as shown in Figure 6(a-d), in accord with the North American monitoring network evaluation shown above, the observed time series in the difference term (blue line, Figure 6(e)) shows that differences between the layer average and 2m ozone values have a significant time variation. The canopy parameterization captures at least some of the broad features of that variation (e.g. compare blue to red line: high values in the difference between July 2nd and July 9th, increases in the difference between July 12 and 14th,

July 23rd to 25th, and July 27th to 29th). The canopy parameterization shows at least some capability to capture the day-to-day variation in the daily differences which make up the average change in the mean bias, albeit at a relatively low correlation coefficient.”

3. Figure 6 – When the authors compare reduction due to canopy models with other reductions of policy-relevant scenarios for climate change, it is once again crucial to clearly discuss the spatial (vertical) extent of these reductions. At the moment, it is not clear to me how to assess these results. It is possible that the reduction associated with climate change occurs in similar proportions within the entire extent of the ABL and that the canopy reductions only occur very close to the ground. This would certainly impact how these results are discussed. More details are needed.

See our above discussion with respect to the new Figure 5. Figure 6 from the previous version becomes Figure 7 in the revised version. All of the values quoted in Figure 7 are from surface concentrations of ozone in references quoted – in that respect, both the references and current work are on an equal footing; an apples to apples comparison of the change in surface ozone concentration. In the revision, we have also shown that the impacts of the canopy reductions, while strongest at the ground, extends throughout the ABL. The papers quoted do not discuss the vertical extent of the changes in as much detail – consequently, we don’t have the means to assess whether the extent to which the ozone changes in the references extend significantly in the vertical. In the revised manuscript, we have, however, shown that the canopy parameterization is not limited to the region immediately close to the surface. A single sentence has been added to the discussion on Figure 7(formerly 6): “We note that these comparisons are for surface ozone changes only – while we have shown in Figure 5 that the impacts of the forest on ozone levels extend throughout the atmospheric boundary layer, we are unable to show similar comparisons for the references quoted.”

4. I find that the manuscript falls short on explaining how the canopy model is actually reducing ozone levels. First, two effects are mentioned: reduction of photolysis rates and modifications in vertical transport. Which one is the most important? How much does each effect contribute to the total changes? Second, how does reduced turbulence diffusivity impacts ozone levels? What is the model actually doing inside the forest? Is there a vertically resolved model for ozone dry deposition and stomata uptake? If the model reduces ozone and I am not told what the physical/chemical process causing the reduction actually are, I am not sure I have learned anything new.

No changes were made to the deposition parameterization between the two simulations – the only differences are the two stated by the reviewer, so they *de facto* account for the differences between the model simulations.

We are glad that the reviewer has asked the question about the relative importance of photolysis versus the turbulence; we examined this in some detail earlier in our development of this work, and this question reminds us to shed some light on this issue.

Our first test at the outset of the work was to determine the potential impact of the photolysis *alone* on the model results. Two simulations of 48 hour forecasts were carried out for the month

of July – the first of these was with the unmodified model, and the second was with a model version in which the gas-phase chemistry solver was called twice at every model time step for the lowest model level: once with the unmodified photolysis rates, and once for the photolysis rates assuming the full attenuation due to the foliage. In other words, the chemistry was solved on a per-time-step basis for conditions above and below the canopy. Once the two chemistry solutions were generated on the given time-step, the results were averaged – that is, under the *de facto* assumption that turbulent mixing within the lowest resolved model layer is almost instantaneous; complete mixing occurring within the time-step. Hence no separation of chemical regimes was allowed to occur for this first test, unlike our subsequent parameterization, with multiple “in canopy” layers and the accompanying within-canopy turbulence parameterization.

Evaluation of this initial test showed that the photolysis rate correction alone was capable of reducing the ozone locally by over 10 ppbv, depending on the time of day, and the North American ozone bias was reduced by 19% (compare to 59% for the combined “v2” model used for the same simulations as the base case). Thus, about 1/3 of the effect can be accounted for by photolysis rates alone, and 2/3 by the turbulence parameterization.

We have included the following text into the revised manuscript, immediately after the statistics reported in Table 1:

“The relative importance of the reduction in photolysis rates versus resolved canopy turbulence was examined in a set of simulations in which only photolysis rates were modified. The gas-phase chemistry of the lowest resolved scale model layer was calculated twice at every time-step in the otherwise unmodified model, once using the original photolysis rates, and once using photolysis rates which had been attenuated with the full LAI depth of the forest canopy. The results of these two lowest layer calculations at each time-step were then averaged, effectively assuming instantaneous mixing between above- and below-canopy conditions. The results of this “photolysis alone” simulation were compared to the base case, and resulted in a relative reduction of ozone mean bias of 19% (compared to 59% for the v2 simulation of Table 1). About 1/3 of the reduction in ozone concentrations may thus be attributed to the reduction in photolysis rates, and a further 2/3 of the reduction may be attributed to the separation in chemical regimes resulting from the turbulence parameterization and additional model layers.”

5. The “parameterization” of the turbulence inside the canopy is anything but elegant. It makes use of a large number of fitted coefficients, with many decimal places, while observational data in canopies shows very large variations from case to case (hardly justifying the choices made by the authors). In particular, the authors point out the importance of LAI, and then use a parameterization that does not even account for the effect of LAI on the turbulence (the effects of LAI on the in-canopy turbulence is one of the very clear effects that the community has been able to establish). Under these conditions, why not adopt a simple model based on physics (such as the Massman and Weil 1999 model, that includes effects of LAI and leaf area density in a simple and elegant way that respects many dynamical constraints) instead of an empirically fit model that has very little physics embedded into it?

We pointed out the importance of LAI with respect to *light attenuation*, but not as a feature in our parameterization of *canopy turbulence*. Rather, we noted that the shape of the profile in Eulerian vertical velocity was sufficiently invariant across the multiple observation studies

collected by Raupach, that relatively simple functional fits to those profiles could be used to describe the generic properties of turbulence and link this to the meteorological model. This approach was followed by Raupach himself in describing generic turbulence profiles within the forest canopy (as was referenced in the original manuscript).

The large number of decimal places are a requirement of the desire to have a smooth transition between different z/h_c levels in the formulae, and should not be misinterpreted as an indication of the accuracy of the parameterization. However, the parameterization has been used in four 1D canopy column simulation publications (three referenced in the original manuscript and an additional paper reference added at this stage of the review process (Ashworth et al, 2015): for each of these papers, this parameterization provided good comparisons to observations of multiple chemical species at multiple heights within and above the forest canopy.

There are many different ways of approaching both the light attenuation and turbulence within a forest canopy in the literature – what we have attempted here is a relatively simple approach, one which has an established track record of providing adequate results from 1D models, has low computational cost, and is relatively easy to implement. We are certain that better approaches are possible. A typical weather forecast or air quality forecast model has multiple options for the processes in the atmosphere – some, like ours, are based on best fits to the available observations, while others are based on underlying physical theory. The constraint determining which parameterization is used is often the computational cost of the numerical methods used and the computational resources available.

At the same time, we do not wish to imply that ours is the only, or even the best, means by which to describe canopy turbulence within the context of a large-scale chemical reaction-transport forecast model. Our hope is that our paper will stimulate new work in this area, bringing together the different disciplines required to generate sufficient observations to test out canopy turbulence parameterizations in more detail.

With regards to Massman and Weil (1999) – we thank the reviewer for this reference, and have added this reference to the revised manuscript as a possible approach to pursue for future improvements for canopy parameterizations. We agree that the inclusion of a leaf-area index term is worthwhile, noting that the Large Eddy Simulation literature quoted in the original manuscript uses an LAI-dependent drag term in an even more “fundamental physics” approach than the parameterization derived in Massman and Weil (1999). However, both Massman and Weil (1999) and our own work suffer from a paucity of observations upon which to base parameterizations. For example, Massman and Weil’s parameterization is based on the assumption of a logarithmic wind profile above the canopy during neutral atmospheric conditions (page 86 of this reference), while the observations of Shaw et al (1988) suggest significant changes in the canopy turbulence behavior between stable and neutral conditions. The Massman and Weil parameterization is dependent on the choice of parameters (α , γ_1 , γ_2 , γ_3 , and A_1) used to fit the parameterization to specific observation datasets. When sufficient information is available to carry out these fits, the performance of the parameterization is impressive (e.g., their Figure 5, a comparison to the observations of Katul and Albertson (1998), where these parameters could be constructed from the observations made in that reference). However, in comparisons where such information was not available, the performance of their

parameterization became unrealistic: in their Figure 8, the observations show values of $\sigma_w(z)/u(z)$ which are either *constant* or *increase* with increasing z/h_c , in contrast to their parameterization, which suggests a strong *decrease* of $\sigma_w(z)/u(z)$ with increasing z/h_c . Massman and Weil (1999) noted for the given comparison that “The model shows a systematic increase in the turbulent intensity with increasing LAI, which was also noted by Cionco (1972) for simple canopies. However, this behavior is not confirmed by the observations in Figure 8, which exhibit more scatter and less systematic trend with LAI than does the model.” Their Figure 3 also suggests to us that the incorporation of LAI is a secondary effect; LAI values of 10.2, 4.0 and 2.9 all giving $\sigma_w(z)/u^*$ values within 0.3 of each other through much of the below-canopy region.

Despite these issues, we feel that this parameterization (and our own!) could be improved with the additional observation data we hope will be collected in response to our work. We have added the following text to the revised manuscript, to the paragraph in the original manuscript between lines 454 and 459: “The dependence of canopy turbulence on leaf area index has also been examined in the literature (Massman and Wei (1999)). When sufficient observation information on specific canopies is available to allow characterization of this methodology’s free parameters, this approach has been shown to provide a good fit to observed canopy turbulence, though performance is degraded in cases of insufficient observation accuracy and for complex canopies. Nevertheless, we feel that this approach and the incorporation of LAI into canopy turbulence parameterizations should be pursued in future work, in conjunction with the collection of additional measurement data for parameterization evaluation.”

6. Line 81 – “suggest that vertical diffusivity in the under-canopy region will be greatly reduced, partially decoupling this region from the rest of the atmosphere, and increasing the relative influence of chemistry on ozone formation and destruction below the foliage.” Please elaborate on how reduced diffusivity would impact these processes. My first reaction is that less ozone would be transported into the deeper layers of the forest, but it is not clear how this would impact ozone formation and destruction.

We’ve expanded this section to better describe the chemistry and its interactions with the turbulence, adding two additional references to Large Eddy Simulations which have recently appeared in the literature, viz:

“...suggest that vertical diffusivity in the under-canopy region will be greatly reduced, partially decoupling this region from the rest of the atmosphere, and increasing the relative influence of chemistry on ozone formation and destruction below the foliage. The reduced diffusivity may slow the upward transport of surface-emitted species such as nitric oxide, and hydrocarbons such as alkenes, increasing their concentrations in the below-canopy space. In a well-lit environment, higher concentrations of these species would lead to ozone *formation* (the dominant reactions being the conversion of nitric oxide to nitrogen dioxide via oxidation reactions, followed by nitrogen dioxide photolysis creating triplet-state monatomic oxygen, in turn biasing the balance of reactions towards ozone formation). However, in the darkened environment below the foliage, the dominant reactions are those of ozone *destruction* through “titration”; the reaction of nitric oxide and/or alkenes with ozone itself. Large eddy simulation studies with passive (i.e. non-reactive) tracers (Queck *et al*, 2014; Kanani-Suhring and Raasch, 2015) have demonstrated

these “trapping” effects of forests, as well as a third important process: species emitted in *non-forested* regions will collect and be trapped in adjacent downwind *forested* regions. This “collection” effect may further enhance the below-canopy pool of precursors to ozone destruction, in the darkened below-canopy environment. These combined effects may result in a shift of photochemical regime away from ozone production and/or towards enhanced ozone destruction.”

We have also added, in the revised discussion following Table 1 of the original manuscript, the following sentence:

“We note that Large Eddy Simulation simulations of non-reactive tracers (Queck *et al*, 2014; Kanani-Suhring and Raasch, 2015)), coupled with our findings, have implications for the monitoring of ozone formation/destruction precursor chemicals: monitoring instrumentation located in clearings or small towns surrounded by forest may report lower concentrations of these species than in the surrounding forests. Monitoring instrumentation at such locations may not be spatially representative of the surrounding terrain.”

New References:

Queck, R., Bernhofer, C., Bienert, A., Eipper, T., Goldber, V., Harmansa, S., Hildebrand, V., Maas, H.-G., Schlegel, F., and Stiller, J., TurbEFA: an interdisciplinary effort to investigate the turbulent flow across a forest clearing, *Meteorologische Zeitschrift*, **23**, 637-659, 2014.

Kanani-Suhring, F., and Raasch, S., Spatial variability of scalar concentrations and fluxes downstream of a clearing-to-forest transition: a Large-Eddy Simulation study, *Boundary-Layer Meteorol*, **155**, 1-27, 2015.

7. Line 191 – “The forest canopy parameterization also improves the root mean square error, mean gross error, coefficient of efficiency, index of agreement and variance.” This is again a bit of an overstatement. I am actually puzzled by how little difference the canopy model makes in all statistics other than the BIAS.

The statistics quoted do improve, as stated, albeit the improvement is not as great as for the bias: usually in the second or third decimal place (bias is in the first decimal place), with the exception of the variance, where the improvement may be seen in the first decimal place (decreasing by almost 1/3) for the first simulation pair, and by 8% in the second simulation pair. The sentence in question has been modified to reflect this more exact description of the model performance: “The forest canopy parameterization also improves the variance (reduced by 1/3 in the v2 simulations and by 8% in the v2.1+ simulations). Some of the other statistics are also improved in the second to third decimal place (root mean square error, mean gross error, coefficient of efficiency and index of agreement).”

There are a host of possible causes for model error – our work has focused on the forest canopy; through improving its treatment within the model we have been able to show a significant improvement in model bias and variance, and lower level improvements in several other

statistics. To the best of our knowledge, no other 3D air-quality model has attempted this level of detail for these processes – and our analysis suggests that the underlying processes have a profound influence on the ozone budget of the lower troposphere. Further improvements to model statistics may be possible with further improvements to the forest parameterization – and/or to all other aspects of chemical reaction transport models - that work is beyond the scope of the current study.

Reviewers' Comments:

Reviewer #4 (Remarks to the Author)

The authors have satisfactorily addressed most of my comments. At least to me, the manuscript is much clearer and the value of the contribution is now obvious. I am ok with the small disagreement left regarding the canopy model. I do believe that the manuscript is suitable for publication in Nature Communications.